# An investigation of unsteady 3D effects on trailing edge flaps

Eva Jost[1], Annette Fischer[1], Galih Bangga[1], Thorsten Lutz[1], and Ewald Krämer[1]

[1]Institute of Aerodynamics and Gas Dynamics, University of Stuttgart, Pfaffenwaldring 21, 70569 Stuttgart, Germany

*Correspondence to:* Eva Jost (e.jost@iag.uni-stuttgart.de)

**Abstract.** The present study investigates the impact of unsteady 3D aerodynamic effects on a wind turbine blade with trailing edge flap by means of Computational Fluid Dynamics (CFD). Harmonic oscillations are simulated on the DTU 10 MW rotor with a morphing flap of 10 % chord extent ranging from 70 % to 80 % blade radius. The deflection frequency is varied in the range between 1p and 6p. To quantify 3D effects, rotor simulations are compared to 2D airfoil computations and the 2D theory by Theodorsen. It was found that the deflection of the flap on the 3D rotor causes a complex wake development and induction which influences the loads over large parts of the blade. Especially, the rotor near wake with its trailing and shed vortex structures revealed a great impact. Trailing vorticity, a 3D phenomenon, is caused by the gradient of bound circulation along the blade span. Shed vorticity originates from the temporal bound circulation gradient and is thus also apparent in 2D. Both lead to an amplitude reduction and shed vorticity additionally to a hysteresis of the lift response with regard to the deflection signal in the flap section. A greater amplitude reduction and a less pronounced hysteresis is observed on the 3D rotor compared to the 2D airfoil case. Blade sections neighboring the flap experience however an opposing impact and hence partly compensate the negative effect of trailing vortices in the flap section with respect to integral loads. Comparisons to steady flap deflections at the 3D rotor revealed the high influence of dynamic inflow effects.

## 1 Introduction

The reduction of ultimate and fatigue loads plays an important role in today's wind energy research. In the background of economic efficiency, load alleviation systems bare potential to reduce rotor weight and costs, to increase the turbine reliability or allow a further enlargement of the rotor radius and thus power output. One promising concept to reduce dynamic load fluctuations are trailing edge flaps applied to the outer part of the rotor blade. As flaps are able to increase or decrease the local lift by adapting the deflection angle, it is possible to partly compensate load variations due to variations of the effective inflow angle and velocity.

Over the last years, several investigations showed the potential of the flap concept as for example a test on a full-scale turbine performed by Castaignet et al. (2014). In aero-elastic simulations, fatigue load reductions up to approximately 30 % have been found for a trailing edge flap covering up to 25 % of the blade span of a 5 MW turbine (Barlas et al., 2012a). In most of the numerical studies the aerodynamic loading was computed by blade element momentum (BEM) codes (e.g., Bernhammer et al., 2016; Chen et al., 2017; Ungurán and Kühn, 2016), which have been extended with different engineering models to account for the unsteady flow (e.g., Bergami and Gaunaa, 2012). As viscous and unsteady aerodynamics have a great influence on

dynamically deflected flaps (Leishman, 1994), it is however important to also apply higher fidelity models and gain knowledge of the flow physics. In this respect, a lot of studies have been performed on 2D airfoils, for example by Troldborg (2005) and Wolff et al. (2014) using CFD. 2D comparisons of simulation methods with different aerodynamic fidelities were performed by Bergami et al. (2015). For the 3D wind turbine rotor, only few publications based on higher fidelity aerodynamic models are available. In 2012 Barlas et al. (2012b) compared CFD to BEM predictions for a rotor with trailing edge flap in an artificial half-wake scenario and found a reasonably good agreement with regard to the complexity of the test case. Leble et al. (2015) investigated trailing edge flaps on the 3D rotor as part of the European AVATAR project and proved the load alleviation potential using a CFD approach. Several comparisons of codes with different aerodynamic fidelities can also be found in the AVATAR project reports (Manolesos and Prospathopoulus, 2015; Ferreira et al., 2015; Aparicio et al., 2016b). A benchmarking within the European Innwind.EU project (Jost et al., 2015a; Barlas et al., 2016) showed however that there are still differences between the results of CFD simulations and BEM methods which need to be analyzed. While a previous investigation focused on the analysis of static flap deflection angles (Jost et al., 2016) by means of CFD, the main objective of the present work is to study the influence of unsteady 3D effects on the example of harmonically oscillating morphing flaps.

Different deflection frequencies ranging from 1p to 6p are analyzed on the DTU 10 MW rotor (Bak et al., 2013) at rated operational condition. These frequencies are considered a realistic operational range for active load alleviation. The investigated flap layout consists of a single morphing flap ranging from 70 % to 80 % blade radius with 10 % local chord extent. This limited dimension along the blade span was chosen to obtain a high impact of 3D effects. In all cases the flap oscillates with an amplitude of $10°$. It shall be noted that the present work does not aim towards an assessment of the flap concept. The objective is to investigate unsteady 3D aerodynamic effects caused by trailing edge flaps and to obtain deeper knowledge about the dominant phenomena as fundamental basis for an enhancement of engineering tools commonly used for load calculations. Within this respect aero-elasticity is not considered since on a flexible blade pitching and plunging movements are superimposed to the flap oscillation and a distinction of the isolated effects would be difficult.

## 2 Aerodynamic effects of trailing edge flaps

### 2.1 2D airfoil

Trailing edge flaps are able to increase or decrease the airfoil lift for respectively positive (downwards) or negative (upwards) deflections due the change of the airfoil camber. As exemplarily displayed in Fig. 1, this leads to a vertical shift of the lift coefficient $c_l$ over angle of attack (AoA) $\alpha$ curve. This possible lift increase is however mostly connected to an increase in drag as depicted in the drag coefficient's $c_d$-$\alpha$ plot. Additionally, the moment coefficient around the quarter chord point $c_m$ is also significantly influenced by the change of the airfoil shape. In general the flap concept aims towards reducing the overall load fluctuation, but in particular the dominant out-of-plane forces and blade root bending moment. They are primarily influenced by the lift coefficient.

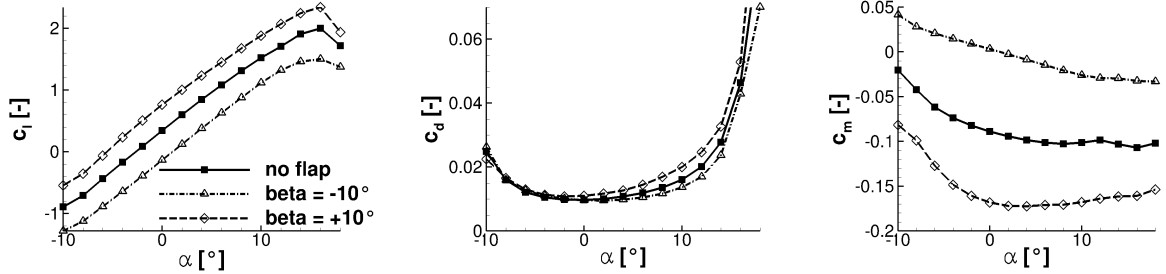

**Figure 1.** Example of flap deflection on $c_l$, $c_d$ and $c_m$ (FFA-w3-241 airfoil, Reynolds number $Re = 15.57e6$, Mach number $M = 0.2$)

## 2.2 3D rotor blade

The increase or decrease of lift in a blade section with trailing edge flap influences the aerodynamic phenomena in most parts of the rotor blade. A qualitative illustration of the vortex development around a rotor blade with deflected flap can be given on the basis of potential flow theory as illustrated in Fig. 2. It shows the vortex system with positive flap deflection in spatial **(a)** and temporal **(b)** consideration.

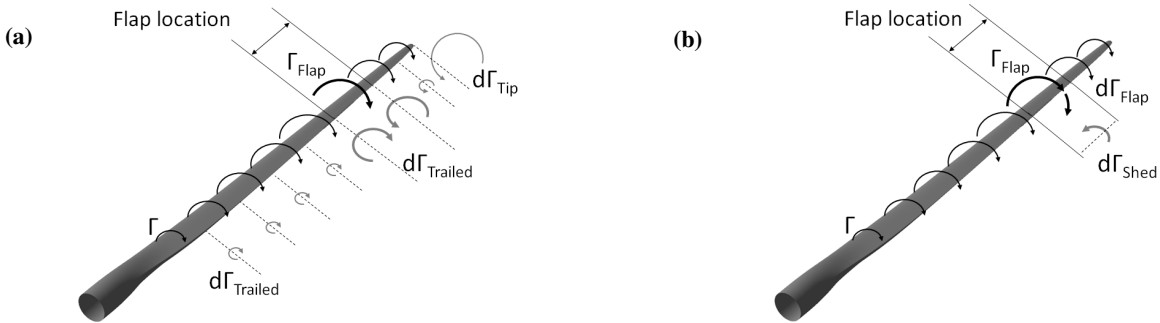

**Figure 2.** Sketch of the bound circulation along a wind turbine blade with trailing **(a)** and shed **(b)** vorticity (part **(a)**: Jost et al., 2017)

Due to the spatial gradient of bound circulation along the blade radius, a vortex sheet trails the rotor blade. In the flap section the bound circulation increases locally due to the change in camber. This leads to higher gradients at the flap edges and hence stronger trailing vortices at these locations. Outboard at the blade tip, the tip vortex is shown. Wake vorticity caused by radially changing bound circulation is commonly referred to as trailed vorticity.

Generally, the efficiency of the flap with regard to local lift increase or decrease is reduced by trailed vorticity in the 3D case. The flap deflection causes an additional downwash or upwash in the flap section. This leads to a respectively lower or higher effective AoA in the 3D case and consequently to induced drag in relation to the baseline AoA. It is worth noting that with respect to the case without flap deflection ($\beta = 0°$), the induced drag is increased in case of positive deflections and decreased in case of negative deflections.

The adverse effect of trailing vortices in the flap section is however countered by a positive effect in the blade parts adjacent to the flap section. Caused by the sign change of induced velocities over the flap edge, the described effects for the flap section are experienced vice versa at these blade parts. With regard to integral loads such as power and thrust, this effect opposes the negative impact of trailing vortices in the flap section.

The temporal consideration (Fig. 2b) displays an increase of bound circulation caused by an increase of the flap angle. This causes shed vorticity with opposed sense of rotation. Shed vortex structures re-induce velocities at the blade location and lead to a change in the effective AoA which in turn influences blade loads. Wake vorticity linked to temporal changes in bound circulation is called shed vorticity.

## 2.3   Theodorsen theory

Shed vorticity has been analyzed by Theodorsen and Garrick (1942) as well as by Leishman (1994) for the 2D case of an airfoil with flap. Theodorsen and Garrick derived an analytical solution for the unsteady airfoil response caused by sinusoidal flap actuation based on his theory from 1935 for thin airfoils (Theodorsen, 1935). This solution is dependent on the reduced frequency (Eq. (1)), one of the most important characteristic parameters when it comes to unsteady aerodynamics (Leishman, 2002). It is a measure of the unsteadiness of a problem as relation between frequency $f$ and chord length $c$ to the inflow

velocity $v_{inf}$.

$$k = \frac{\pi \cdot f \cdot c}{v_{inf}} \tag{1}$$

According to Theodorsen's method the lift is given by:

$$c_l(t) = \underbrace{2\pi C(k) \left( \frac{F_{10}\beta}{\pi} + \frac{F_{11}\dot{\beta}c}{4\pi v_{inf}} \right)}_{1} + \underbrace{\frac{c}{2v_{inf}^2} \left( -v_{inf} F_4 \dot{\beta} - \frac{c}{2} F_1 \ddot{\beta} \right)}_{2} \tag{2}$$

The derived function consists of two terms: a first term which represents the circulatory forces connected to the bound

circulation, and a second term which accounts for added mass effects. This second term is mostly referred to as non-circulatory part and depicts the influences of the inertia of the fluid. In this function, $\beta$ represents the instantaneous flap angle and its time derivatives. The coefficients $F_1$ to $F_{11}$ are geometric terms solely dependent on the relative flap length to chord. For their definition it shall be referred to Leishman (1994). The function $C(k)$ is the complex Theodorsen function which accounts for the effects of the shed wake.

The instantaneous lift coefficient $c_l(t)$ can be analyzed with regard to the amplitude $\Delta c_l$ and phase shift $\phi$ of the lift response with respect to the input flap signal. $\Delta c_l$ can also be evaluated in relation to the amplitude of flap deflection ($\Delta c_l / \Delta \beta$). Figure 3 shows the solution for a flap length of 10 % chord as function of the reduced frequency $k$. In the diagram two curves are plotted: a solid curve which represents the solution of the complete function $c_l(t)$, and a dashed curve which shows the solution if only the circulatory components are regarded and apparent mass effects are neglected. No major difference between both curves

is apparent in $\Delta c_l/\Delta\beta$. It is continuously decreasing with $k$ in the displayed range which is applicable for this work as the investigated flap frequencies correspond to reduced frequencies of $k = 0.024 - 0.147$ at mid flap position. An increasing lag can be observed in phase shift, which is more pronounced if only circulatory components are included. While below a value of roughly 0.1 the differences between both curves are small, higher discrepancies are observed at larger $k$ when apparent mass effects become increasingly dominant. In conclusion it is however found that the non-circulatory contribution is in the investigated range of reduced frequency of minor importance.

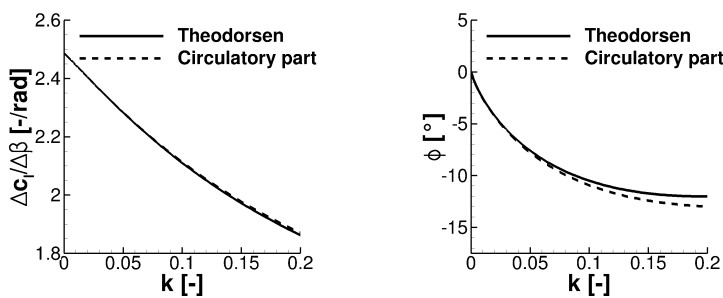

**Figure 3.** Lift amplitude in relation to flap amplitude $\Delta c_l/\Delta\beta$ and phase shift according to Theodorsen

Additionally to the lift equation, Theodorsen developed functions for $c_m$, the pressure drag $c_{dp}$ and the flap hinge coefficient $c_{m,fh}$. It is referred to Hariharan and Leishman (1996) for their definition. Theodorsen's derivations include the assumptions of thin airfoils and 2D flow. Both are not applicable for the aerodynamics of a modern wind turbine since current developments in blade design tend towards thicker airfoils for increased stiffness. But since the theory is well-known and commonly used to determine unsteady aerodynamic characteristics, it is compared to the obtained results.

## 3 Numerical approach

### 3.1 Simulation process chain

The process chain for the simulation of wind turbines, which was developed at the Institute of Aerodynamics and Gas Dynamics (IAG), University of Stuttgart (Schulz et al., 2016), is used in the present work. The main part constitutes the CFD code FLOWer of the German Aerospace Center (DLR) (Kroll and Fassbender, 2002).

FLOWer is a compressible code that solves the 3D Reynolds-Averaged Navier-Stokes (RANS) equations in integral form. The finite-volume numerical scheme is formulated for block-structured grids. A second order central discretization with artificial damping is used to determine the convective fluxes, which is also called the Jameson-Schmidt-Turkel (JST) method. Transient simulations make use of the implicit Dual-Time-Stepping scheme. To close the RANS equation system, several state-of-the-art turbulence models can be applied, as for example the SST model by Menter (1994) used in this study. FLOWer offers the use of the CHIMERA technique for overlapping meshes which is applied in the simulation of 3D wind turbines.

Grid generation is widely automated with scripts for 2D airfoils and 3D rotor blades. The generation of the blade grid for example is conducted with Automesh, a script developed at the IAG for the commercial grid generator Gridgen by Pointwise. The blade grids are of C-type with a tip block and coning towards the blade root in order to connect to the turbine spinner. Spinner and nacelle are typically included in the simulations. In case of pure rotor simulations as performed in this study, the computational domain is modeled as a 120°-model with periodic boundary conditions on each side to reduce computing efforts. For the present study, this means that the flaps of all blades are deflected simultaneously.

On the post-processing side, again several scripts are available for the analysis of the simulations. Loads are calculated by the integration of pressure and friction distribution over the blade surface. Sectional distributions along the blade radius are determined similarly by dividing the blade into different radial sections.

## 3.2 Trailing edge flap model

Trailing edge flaps are modeled based on grid deformation in FLOWer. Therefore, the deformation module (Schuff et al., 2014) was extended by a polynomial function (Daynes and Weaver, 2012; Madsen et al., 2010) to describe the shape of the deflected flap.

$$w = \varphi(x) \cdot \beta \quad \varphi(x) = \begin{cases} 0 & 0 \leq x < (c-b) \\ \frac{(c-x-b)^n}{b^{n-1}} & (c-b) \leq x \leq c \end{cases} \tag{3}$$

In Eq. (3) $c$ represents the chord length and $b$ the flap length. The result $w$ is the vertical change in y-direction, while the movement in x-direction is neglected for small deflection angles up to 10°. Using this function requires the chord to be aligned with the x-axis. The polynomial order $n$ is set to 2 for this investigation. In Fig. 4 the deformation methodology shown for a 2D airfoil section. The un-deformed and deformed airfoil surface are shown serving as input to the grid deformation algorithm, which computes the new simulation grid at each time step.

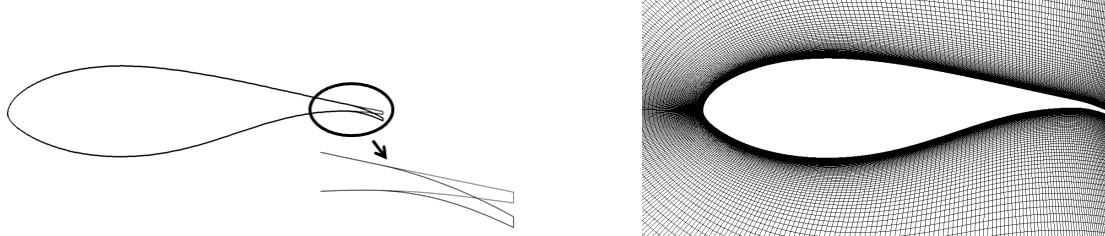

**Figure 4.** Methodology for 2D deflection (Jost et al., 2016)

The approach for the blade mesh is displayed in Fig. 5. There is no separate grid for the flap part. It is integrated into the blade grid. The connection between the moving flap part and the remaining rigid blade surface is computed by the deformation

algorithm which generates a smooth transition. At the location of these transitions the blade grid is radially refined to capture gradients in the flow field which are expected to occur.

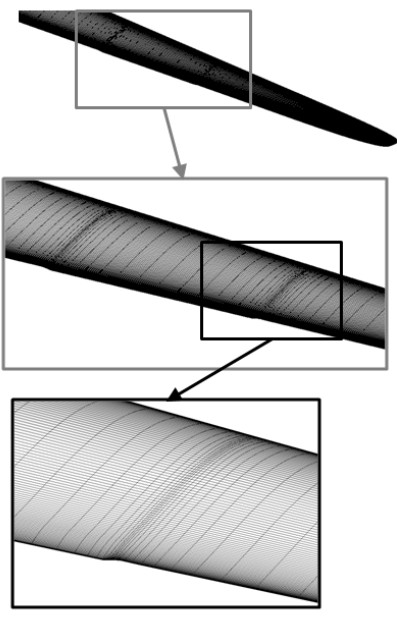

**Figure 5.** Methodology for 3D deflection

### 3.3 Code-to-code validation of the simulations

A baseline simulation setup for the DTU 10 MW turbine without flap has been examined and validated by code-to-code
5   comparison within the AVATAR project. For this purpose a simulation of the power curve on the basis of the stiff straight blade without precone was conducted in steady mode. The comparisons between the different codes of the project partners is presented by Sørensen et al. (2015). The FLOWer results showed a good accordance with results obtained with EllipSys3D by DTU and MapFLow by NTUA. A detailed analysis of the FLOWer results with a special focus on the comparison of steady and unsteady simulations is performed in Jost et al. (2015b). A comparison of the simulations with flaps to the AVATAR project
10  partners can be found in Ferreira et al. (2015) and Aparicio et al. (2016b).

### 3.4 Grid generation

For the 2D airfoil simulations, the 75 % blade cut of the DTU 10 MW turbine (FFA-w3-241 airfoil), which is the mid flap position in the chosen trailing edge flap configuration, was extracted from the geometry. The airfoil grid was generated using a script for the commercial grid generator Pointwise. Approximately 180,000 grid cells have been used with 417 surface nodes

and 205 nodes in wall-normal direction. Boundary layer resolution was chosen for $y^+_{max} \sim 1$ and the farfield boundary is located at 150 chords distance.

For the rotor simulations, the setup used in the code-to-code validation was modified in order to simulate the rotor with trailing edge flap. Flap edge refinements were included into the blade grid and a higher resolved background grid was chosen to accurately capture wake effects. A separate grid convergence study of the blade grid was performed with a steady flap deflection +/- 10° and the results are presented in Jost et al. (2015a). The final setup used in the present study amounts approximately 21.65 million cells. The distribution between the different grids is shown in Table 1.

**Table 1.** Grid cells in the 120°-model

| Blade | Spinner | Nacelle | Background | Total |
|-------|---------|---------|-----------|-------|
| 8.16e6 | 1.39e6 | 1.45e6 | 10.65e6 | 21.65e6 |

## 3.5 Temporal discretization of the simulation setup with flaps

Another critical issue with regard to the unsteady simulations is the temporal resolution meaning the choice of time step. Unsteady simulations in FLOWer make use of the Dual-Time-Stepping method as implicit scheme. In this approach a pseudo-time is introduced into the equation system at each time step for which a steady solution is obtained. The method allows the choice of significantly larger time steps than those dictated by the CFL condition in explicit schemes. However, the actual eligible size is problem dependent. In most cases the time step is a trade-off between simulation accuracy and computational time and it is necessary to determine the largest possible time step that can still resolve the unsteady flow effects sufficiently. To analyze the influence of the temporal discretization within this study, a sensitivity study has been performed based on 2D and 3D simulations

At first, the 2D airfoil case (FFA-w3-241, 75 % blade radius) is presented. The simulations have been performed at a realistic inflow extracted from the 3D rotor case at rated operational conditions. These conditions are specified in Table 2.

**Table 2.** DTU 10 MW turbine, rated operational conditions

| Wind speed | Rotational speed | Blade pitch | Tip speed ratio |
|-----------|------------------|-------------|-----------------|
| 11.4 m/s | 9.6 rpm | 0° | 7.86 |

At 75 % radius, the Reynolds number was determined to 15.4 millions, Mach number to 0.2 and the AoA to 6.5°. For the determination of the time step influence, the high flap frequency corresponding to six times the rotational frequency at rated operational point (6p) was chosen, which corresponds to a reduced frequency of 0.147. Results for $c_l$ and $c_d$ are shown in Fig. 6. As similar findings were made for $c_m$, it is not separately shown here but will be regarded in 3D. Four different time steps have been selected for the study and 100 inner iterations are performed in the Dual-Time-Stepping scheme for all investigated time

steps except the small step of 0.028° for which 30 inner iterations are regarded sufficient. As the results are transferred to the 3D case later on, the different time steps are designated by the corresponding azimuth step in a rotor simulation at rated rotational speed. A time step of 0.028° is for example equivalent to 4.8e-4 s. This very small step correlates to 100 steps per convective time unit, which is in this case the chord length. With regard to the computational effort this time step is not realizable for the 3D case, but serves as reference in this 2D study. All other discretizations are applicable for pure rotor simulations. In both aerodynamic coefficients the influence of the time step size is apparent, but the effect on drag is slightly more distinct. In general, lift and drag agree well for all resolutions except for 1° where major differences are observed.

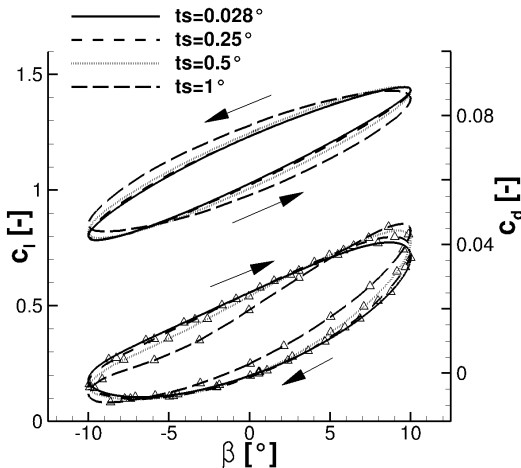

**Figure 6.** 2D influence of time step, lift (no symbol) and drag (symbol △)

Another parameter of influence is the amount of inner iterations in the Dual-Time-Stepping scheme which is analyzed for the time step of 0.5° with three different amounts of inner iterations, 50, 100 and 200. The results are displayed in Fig. 7. Again the results for the small time step of 0.028° is shown as reference.

Generally, it is observed that the temporal accuracy is more dependent on the total amount of iterations per convective unit, than the choice of time step or inner iterations. By comparing the plots in Fig. 6 and Fig. 7, it can be seen that for example the hysteresis for 1° and 100 inner iterations is very similar to the curve with 0.5° and 50 inner iterations. This conclusion can however not be transferred to separated flows, for which a small time step is needed to resolve effects correctly.

Based on these outcomes, simulations of the rotor model have been conducted in order to get an impression of the 3D case. The flap is again oscillating with 6p frequency at rated operational condition. Similar time step sizes as in the 2D case have been chosen replacing 0.028° with 0.125° as further halving. 100 inner iterations are used. Figure 8 shows the resulting driving force, thrust and torsion moment variations at mid flap position. The torsion moment is evaluated relative to the blade pitch axis and positive reducing the blade pitch. The forces and moment are normalized with the total mean value to allow an easier assessment of the differences. While thrust and torsion moment show a good agreement for nearly all time step sizes, higher deviations are observed in the driving component in which the drag differences have a stronger impact. However, a

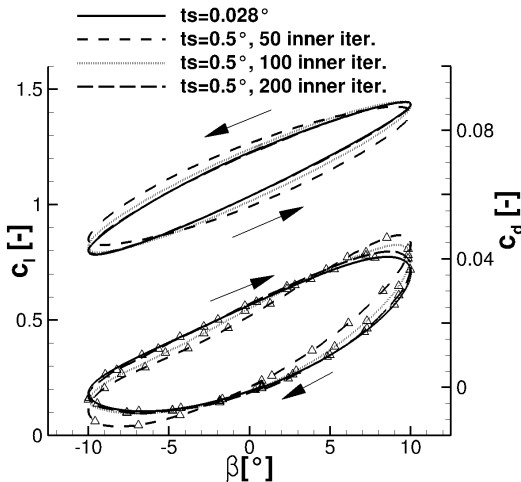

**Figure 7.** 2D influence of inner iterations, lift (no symbol) and drag (symbol △)

convergence of the curve progressions with decreasing time step size can be observed leading to small differences between $0.125°$ and $0.25°$.

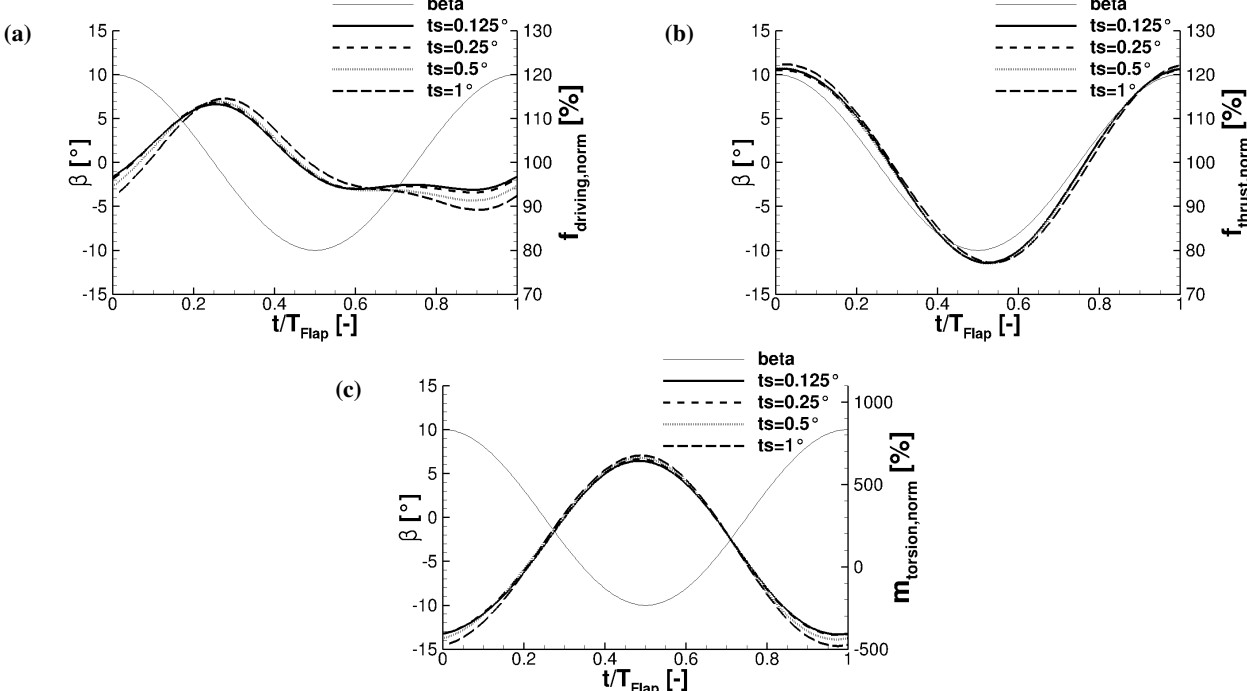

**Figure 8.** 3D influence of time step, 6p, 75 % blade cut, normalized driving force **(a)**, thrust **(b)** and torsion moment **(c)**

To conclude the temporal discretization study, a time step size of 0.25° with 100 inner iterations for a flap frequency 6p shows sufficient accuracy in 3D simulations as trade off to computational time. This corresponds to 240 steps per flap oscillation. In 2D simulations smaller time steps correlated to the convective unit are feasible and consequently used.

## 4   Results

### 4.1   3D rotor simulations with oscillating flap

At first, results of the different flap frequencies are compared for the 3D rotor simulations. All simulations have been conducted at rated conditions (see Table 2). As ambient conditions an air density of 1.225 kg/m$^3$ and a temperature of 288.15 K are used. The simulations were started as steady state computation on two multi grid levels with 8000 iterations respectively and a flap angle of 0°. This steady solution is then restarted in unsteady mode and simulated for the amount of revolutions required for converged loads. With regard to the Menter SST turbulence model, it has to be mentioned that the required wall distances of each cell were computed only once at the simulation start and not updated in every time step. This was done since only minor influence was found in 2D airfoil simulations and to save computation time as in 3D the wall distance calculation is very time consuming. Please note that while in the previous time step study a cosine function is used as deflection signal, now a sinus function is applied (Eq. (4)).

$$\beta(t) = 10° \cdot \sin\left(2\pi t \frac{N}{T_{Rotor}}\right) \qquad N = [1,2,3,6],\ T_{Rotor} = 6.25s \qquad (4)$$

The flap frequencies correspond to reduced frequencies of $k = 0.024(1p) - 0.147(6p)$ at mid flap position. Figure 9 shows the results of integral power and thrust plotted over one rotor revolution. The effect of the flap can be seen clearly in both diagrams. Power and thrust are oscillating with the respective frequency. A higher frequency fluctuation is also apparent in the graphs, which results from unsteady flow separation at the cylindrical blade root. As illustrated in the streamlines plot on the blade surface shown in Fig. 10, flow separation is dominant there. Due to this superposition of effects in the integral forces, it is necessary to regard sectional forces at a blade cut belonging to the flap part in order to investigate the flap effects.

In the following, like in the time step study, the 75 % cut as mid flap position was extracted from the simulations. Figure 11 shows the results of the local driving force, thrust and torsion moment at this location over one flap period for all simulated frequencies. Thrust shows the expected amplitude decrease and phase shifts with increasing frequency. A significant reduction of the amplitude is observed between the 2p and 3p case with 3p and 6p showing a very similar curve progression. These phenomena will be analyzed in more detail in Sect. 4.4. Larger differences are observed in the driving component. For the 3p and 6p case, a second superimposed oscillation is visible from $t/T_{Flap} \approx 0 - 0.2$ and $t/T_{Flap} \approx 0.8 - 1$. This oscillation results from the overlay of lift and drag forces in the rotor plane. At higher frequencies drag shows a significant amplitude increase as seen in Fig. 6 and, additionally, $c_l$ and $c_d$ are oscillating with different phases. The superposition of both force components leads to the curve progression seen in the 3p and 6p case. This phenomenon in driving force is especially present at operational conditions with a pitch angle of 0°, for which the impact of drag is high. Please note that in the plot of the local

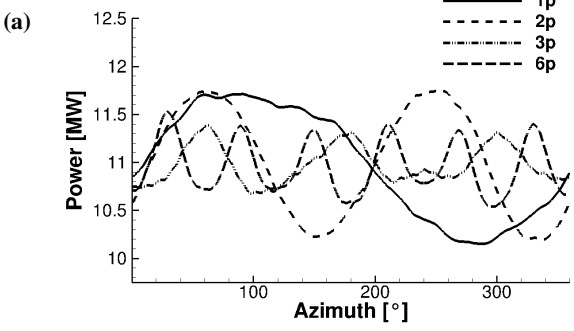

**Figure 9.** Integral rotor power **(a)** and thrust **(b)**

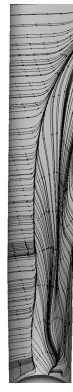

**Figure 10.** Blade root flow separation - pressure side

torsion moment (Fig. 11c), the $\beta$-axis is reversed in order to be able to better judge the phase shift. The moment shows an increase in amplitude with increasing flap frequency, mainly between the 2p and 3p case like it was observed vice versa for thrust. The high frequencies 3p and 6p are again similar. All curves show a lead with respect to the flap signal. A more detailed elaboration of the torsion moment relative to the quarter chord point is also performed in Sect. 4.4.

5      In Fig. 12 and Fig. 13 sectional distributions of driving force, thrust and torsion moment over the blade radius are shown for 1p and 6p case respectively. Four instantaneous solutions are plotted for maximum, minimum and $0°$ flap deflection. Thrust shows the expected increase and decrease in the flap section with a smooth load distribution over the flap edges. This smoothing is a consequence of the positive effect of the flap deflection on neighboring blade sections as described in Sect. 2.2. While trailing vorticity reduces the effect of the flap in the flap section compared to 2D, the sections next to the flap part

10   produce higher/lower lift due to the induced upwash/downwash for respectively positive/negative flap angles. The change of sign in induced velocity caused by the flap edge vortices is also apparent in the driving force as significant steps are appearing at the transition between flap and rigid rotor part. An opposite behavior of sectional driving force in relation to thrust can be noticed by comparing the diagrams. When thrust increases locally in the flap area, the driving force decreases in relation to neighboring sections. This results again from the strong influence of drag on the driving component at rated wind turbine

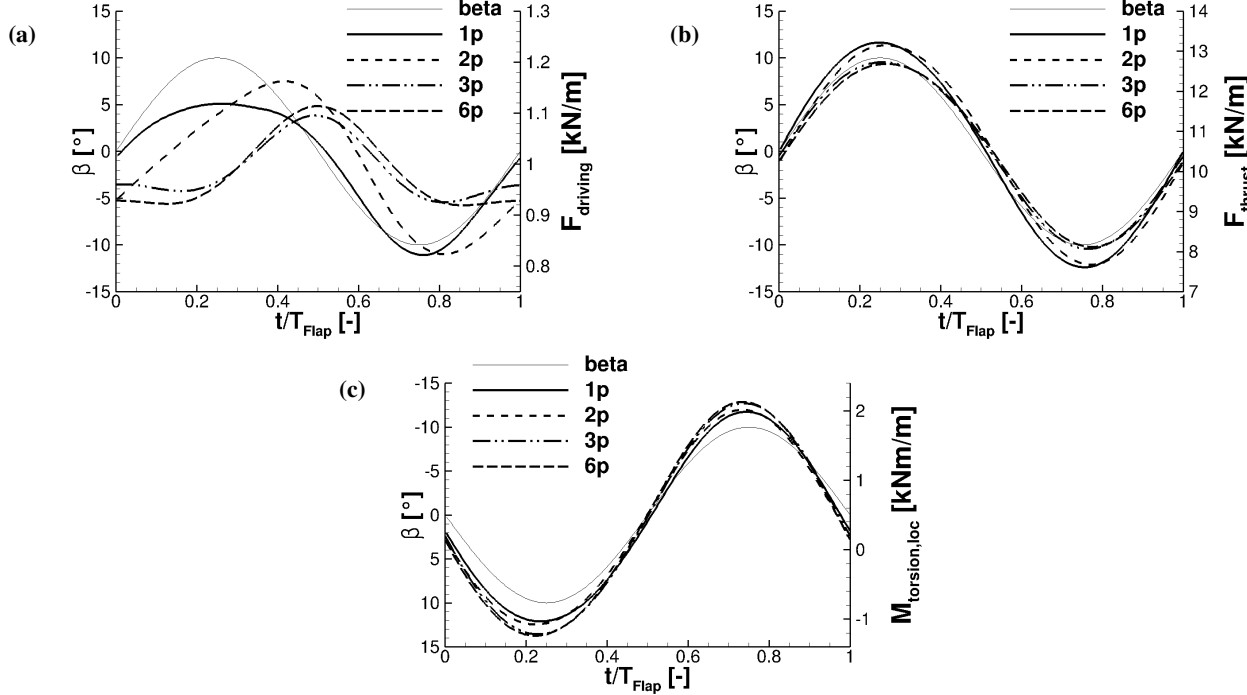

**Figure 11.** Variation of flap frequency, 75 % blade cut, driving force **(a)**, thrust **(b)** and local torsion moment **(c)**

conditions and will be explained on the basis of 1p case as follows. Due to the low reduced frequency in this case ($k = 0.024$), the influence of shed vorticity is still weak. For maximum positive deflection ($t/T_{Flap} = 0.25$) the increase of trailing vorticity causes a downwash in the flap section. This reduces the effective AoA and leads to a rise of induced drag in addition to the drag augmentation caused by the flap deflection itself. The overall drag increase is compensated by the lift increase resulting

5   from the flap deflection and relative to 0° flap deflection an increase of driving force is achieved. The neighboring sections to the flap experience an additional upwash in case of positive deflections. Consequently, the induced drag reduces associated with the lift increase and these sections produce in total a higher sectional driving force. Similar observations are made vice versa for maximum negative deflections, but the driving force increase in the flap section is less pronounced compared to the decrease in case of positive deflection. Further elaborations in this respect can be found in Sect. 4.3, in which lift and drag

10   forces are extracted and compared. With regard to the torsion moment around the pitch axis, a strong oscillation is seen in flap section with steep gradients at the flap edges. This torsion moment or $c_m$ oscillation is typical for trailing edge flaps (Ferreira et al., 2015) and its effect on the overall performance of the flap concept needs to be investigated separately in an aero-elastic simulation when the blade is able to twist.

The differences caused by unsteady effects can also be observed in the plots. Larger variations of the forces are seen in the 1p

15   case compared to the 6p case over the whole blade part influenced by the flap. In contrast the variation of the moment is slightly increasing. The hysteresis can especially be noticed at the time instance when the flap is positioned at 0° deflection. While in

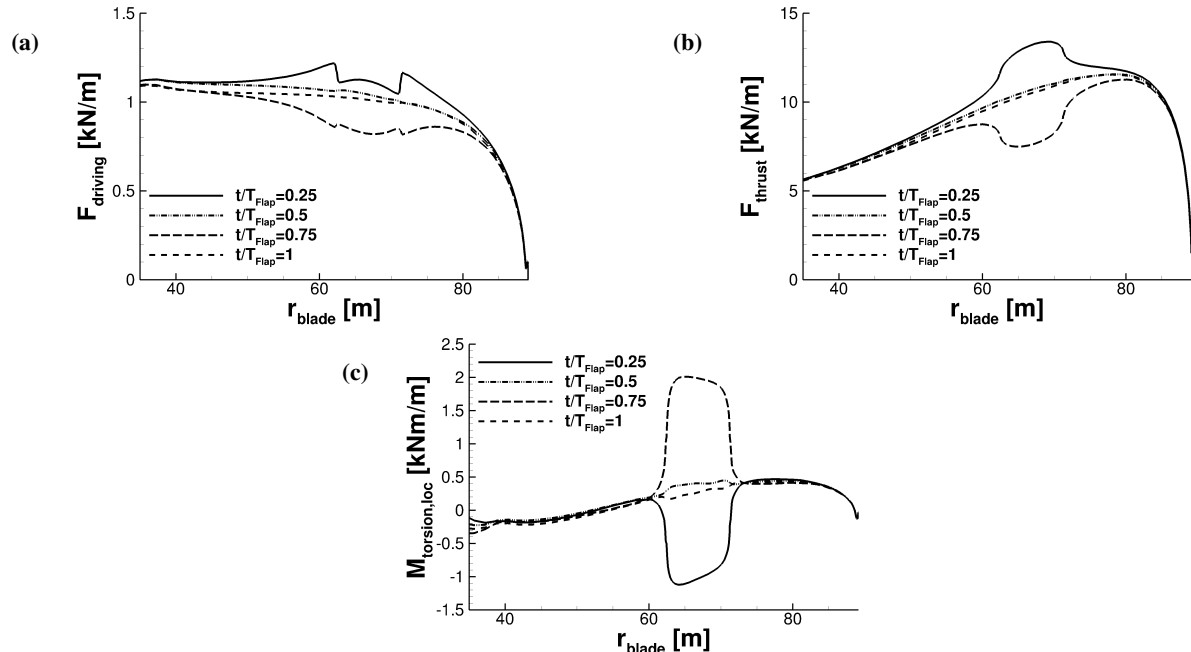

**Figure 12.** 1p sectional forces (driving force **(a)**, thrust **(b)** and local torsion moment **(c)**)

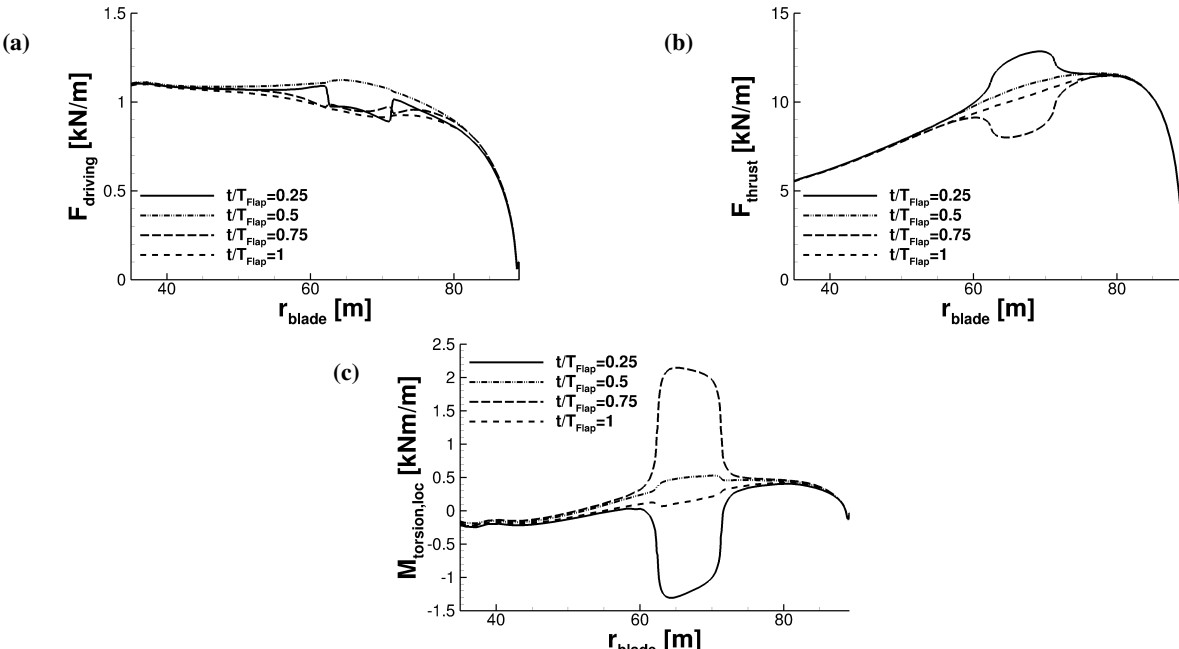

**Figure 13.** 6p sectional forces (driving force **(a)**, thrust **(b)** and local torsion moment **(c)**)

the 1p case the sectional loads at increasing or decreasing flap angle are closely together, in the 6p case larger differences are seen. For the decreasing flap angle at $t/T_{Flap} = 0.5$ the loads are higher than for the increasing flap angle at $t/T_{Flap} = 1$.

## 4.2 Comparison to steady flap deflections

In a first step to analyze the influence of unsteady effects in 3D, the results are compared to the simulations of steady flap deflection for +/-10°. Like for the oscillating flap cases, the simulations were initiated in steady state with 16000 iterations and then restarted in unsteady mode for three turbine revolutions with a time step corresponding to 2° azimuth. This approach is plausible because the flap area is characterized by a steady flow situation as shown in similar studies (Aparicio et al., 2016b). Table 3 shows the mean integral loads of the third rotor revolution normalized with the respective value for $\beta = 0°$ for a relative comparison.

**Table 3.** Integral loads for steady deflection normalized with value for $\beta = 0°$

|  | $\beta = 10°$ | $\beta = -10°$ |
| --- | --- | --- |
| Normalized power [%] | 96.2 | 98.3 |
| Normalized thrust [%] | 103 | 95.2 |

By comparing to the results of the oscillating flap (Fig. 9), it can be noticed that in the oscillating cases an increase of power is possible, while for steady deflections this is not the case. A negative flap angle even leads to a higher power output than a positive deflection. This phenomenon is caused by the differences in axial induction. For positive deflections $c_l$ increases, more energy is extracted from the wind and consequently the axial inductions also rises. This leads to a lower AoA at the rotor blades, which reverses the effect of the flap with regard to power. The opposite is observed in case of negative flap angles. But since less energy is extracted from the wind, still a lower power output compared to the neutral flap case is observed. In this background thrust also shows reasonable values. The magnitudes in the steady cases are lower compared to the 1p oscillating case, for which the axial induction is not able to fully adjust to the changed load situation and consequently higher oscillations are possible.

To verify this hypothesis an extraction of the local AoA along the blade radius has been performed according the reduced axial velocity method (Johansen and Sørensen, 2004). This method has proven to produce reasonable results (Bangga et al., 2016; Klein et al., 2014), but is only applicable for steady inflow conditions. But since in the 1p case the reduced frequency is still very low with a value of 0.024 at mid flap position, a quasi-steady approach is appropriate (Leishman, 2006). The method requires annular elements at different radii in front and behind the rotor plane as input. These elements are placed at an axial distance of one local chord to the rotor blade for the present evaluation. The choice of this axial distance has however shown to have an influence on the results with maximum discrepancies of about 0.16° when the axial distance is reduced for example to 0.2 local chords. Nevertheless, the results of the different cases can be compared to each other and give a qualitative and within this tolerance quantitative analysis. The plot displayed in Fig. 14 highlights the differences for steady and oscillating

flap deflections and underlines that in unsteady cases the axial induction is only slowly adjusting. Consequently dynamic inflow effects play an important role on trailing edge flaps, especially at lower flap frequencies.

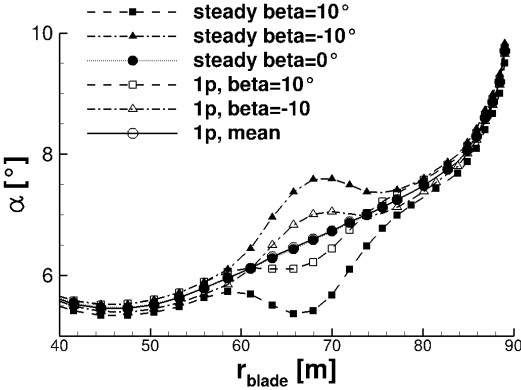

**Figure 14.** Extracted AoA according to Johansen and Sørensen (2004)

Figure 15 and Fig. 16 show extracts from the flow field at an axial distance of one local chord in front and behind the rotor blade for respectively maximum positive and flap negative angle.

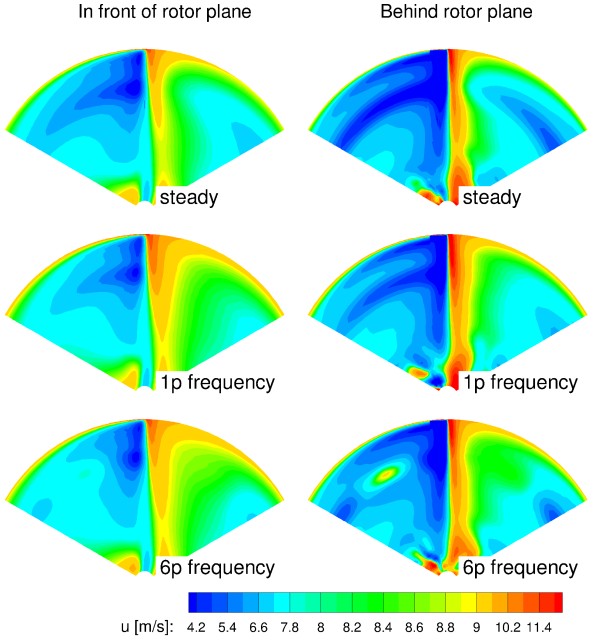

**Figure 15.** Flow field extracted at one local chord distance in front and behind the rotor blade for $\beta = 10°$

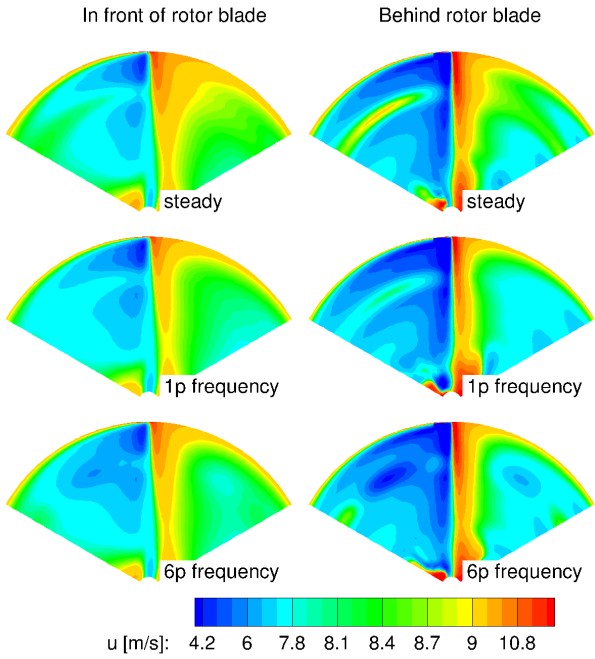

**Figure 16.** Flow field extracted at one local chord distance in front and behind the rotor blade for $\beta = -10°$

The contour of the axial velocity $u$ is displayed from a front view to the turbine which means that the rotational direction is clockwise. The rotor blade is positioned upright. In each figure three cases are shown, steady flap deflection, 1p oscillation and 6p oscillation. Clearly the flow acceleration towards the blade and the reduction of the wind speed in the blade wake can be seen in all cases. In the flap section the opposing deflection can mainly be identified in the blade wake, where less reduction is observed for a negative angle and a higher reduction for a positive angle. By comparing steady deflections to the 1p frequency, the different axial induction can be seen. When comparing the 1p and 6p frequency, the different flap frequencies can also be noticed in the blade wake. In the 6p case at for example $\beta = 10°$, an area with increased velocity is apparent.

### 4.3 Influence of varying AoA in 3D

As observed in the previous section, in the 3D rotor case the local AoA is oscillating over a flap period as a result of dynamic inflow. This means from an aerodynamic point of view that two unsteady mechanisms are superimposed: pitch and flap oscillation. The objective of the present work is though to characterize and quantify unsteady 3D effects solely due to flap deflection and consequently some preliminary considerations have to be made. For this purpose the 1p frequency is regarded in the following, for which quasi-steady assumptions are eligible and the reduced axial velocity method can be applied. The variation of the local inflow velocity and the AoA is shown in Fig. 17 for the mid flap position. While the inflow velocity shows no major variations, the AoA oscillates with an amplitude of $0.6°$.

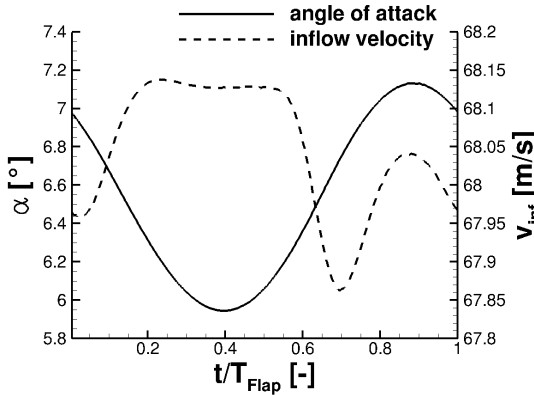

**Figure 17.** 1p instantaneous inflow conditions 3D, 75 % radius

When the instantaneous AoA is extracted from the 3D simulation it includes the oscillations caused by both mechanisms, dynamic inflow and flap oscillation. The dynamic inflow oscillation represents an oscillation of the baseline AoA as a result of the variation of the axial induction of the turbine. In contrast the oscillations caused by the flap originate from the downwash of 3D trailing vorticity which changes the effective AoA. As the objective is to quantify 3D trailing vorticity, the flap-caused
AoA oscillation should be mimicked to the aerodynamic coefficients, while in theory the influence of the dynamic inflow caused oscillation should be eliminated. A clear distinction between both oscillations is however not possible and requires further aerodynamic modelling. Nevertheless, the influence of the overall AoA oscillation on the 3D extracted aerodynamic coefficients can be assessed.

Figure 18 presents the resulting $c_l$ and $c_d$ variations in addition to the resulting variations for an averaged AoA of 6.5° and
local inflow velocity of 68 m/s ($c_{l,mean}$, $c_{d,mean}$). The moment coefficient $c_m$ is not dependent on the inflow direction so that the evaluation of the AoA uncertainty for the $c_m$ behavior can be excluded in this section. It can be seen that the AoA oscillations have only a minor influence on the value of $c_l$, but a strong impact on $c_d$. This is reasonable as for the determination of $c_l$ and $c_d$ in the 3D case, the forces are integrated from the surface solution as driving force and thrust components at first and then transferred to the local inflow or also called aerodynamic coordinate system. The procedure is shown in Fig. 19 for
both components. To determine total lift and drag forces both shares by driving force and thrust are summed up. As the value of $c_l$ is roughly 100 times larger compared to $c_d$, a projection difference of 0.6° as observed in the 1p case has only a negligible impact on $c_l$. Consequently, the results for $c_l$ and $c_{l,mean}$ are very similar. $c_d$ and $c_{d,mean}$ differ however strongly. Based on the previous considerations this difference consists of two parts, induced drag which originates from trailed vorticity and the drag resulting from the AoA oscillation caused by dynamic inflow.

The plot in Fig. 18 can be directly linked to the curves for driving force and thrust in the 1p case displayed Fig. 11 and Fig. 12. Thrust is dominated by $c_l$ and consequently the progressions over a flap period are very similar. The driving component is a superposition of both forces which are oscillating at a different phase. This can be noticed for example at the time instance when

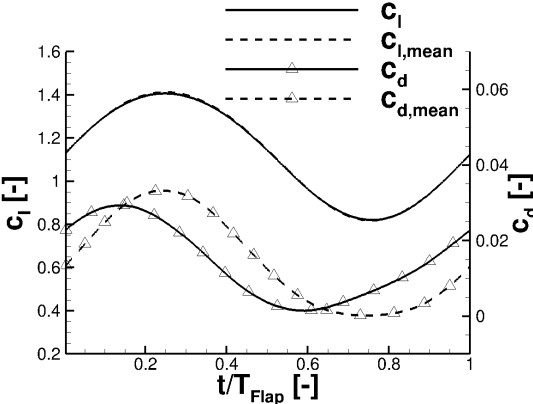

**Figure 18.** 1p comparison of lift and drag, 3D instantaneous/ 3D mean, 75 % radius

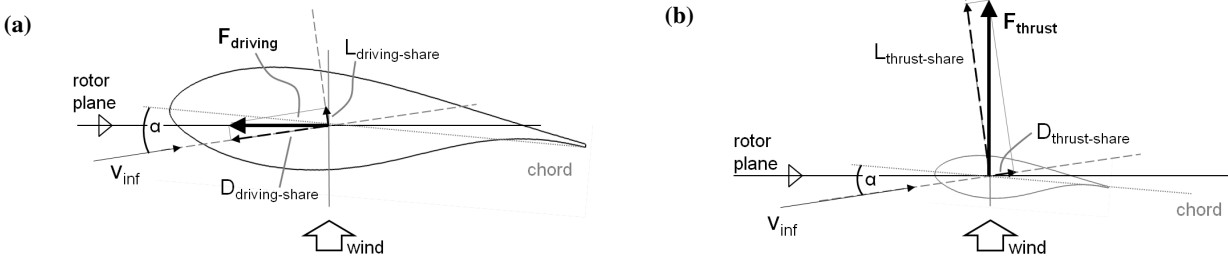

**Figure 19.** Transformation of driving force and thrust to local aerodynamic coordinate system (driving force **(a)** and thrust **(b)**)

the flap is deployed to maximum deflection ($t/T_{Flap} = 0.25$). $c_l$ but also $c_{d,mean}$ is maximal and as a result the progression of driving force flattens. Vice versa the phenomenon is observed at minimum deflection ($t/T_{Flap} = 0.75$).

With regard to the objective of this section, determining the influence of the AoA oscillation on the 3D extracted aerodynamic coefficients, it can be concluded that the extracted $c_l$ is only minor affected and it is eligible to use $c_{l,mean}$ for the comparison

5    to 2D simulations and the evaluation of the impact of trailing vortices. With respect to $c_d$ it is difficult to clearly distinguish between the part caused by the AoA oscillation and the induced drag from trailed vorticity. In order to judge the impact of trailing vortices on $c_d$, this differentiation is however necessary and the part caused by the AoA oscillation needs to be eliminated. The emphasis of the comparison to 2D simulations is hence on lift and moment coefficient.

In order to quantify the limitations of the average approach with respect to the phase shift, an FFT analysis of the results

10   has been performed for the 1p frequency and the main peak in frequency was analyzed. The averaged and instantaneous solution differ by only 1 % in lift amplitude and $0.26°$ in phase shift, which is below the time step resolution of $1.5°$. Due to the limitations caused by the time step size hysteresis effects are only qualitatively judged by comparing the curves, lift amplitudes, however, can be opposed quantitatively.

### 4.4 Comparison to 2D simulations

To study the unsteady phenomena in more detail and to analyze the main effects in the 3D case, the instantaneous $c_{l,mean}$ and $c_m$ results of the 3D extraction are compared to 2D airfoil simulations of the mid flap position at mean inflow conditions. The mean inflow conditions used as input for all flap frequencies in 2D are again: $AoA = 6.5°$ and $v_{inf} = 68m/s$. This leads to a Mach number of 0.2 and a Reynolds number of 15.4 million.

Results of the $c_l$ comparison between 2D static deflection, 2D sinusoidal deflection and 3D sinusoidal deflection are shown in Fig. 20. In order to get an impression of the influence of AoA oscillation in the 1p case, additionally an AoA corrected version of the 2D sinusoidal oscillation case is plotted which is computed by Eq. (5).

$$c_{l,2Dsinus,AoAcorr}(t) = c_{l,2Dsinus}(t) + 2\pi \left( \alpha_{3Dsinus}(t) - \alpha_{mean} \right) \tag{5}$$

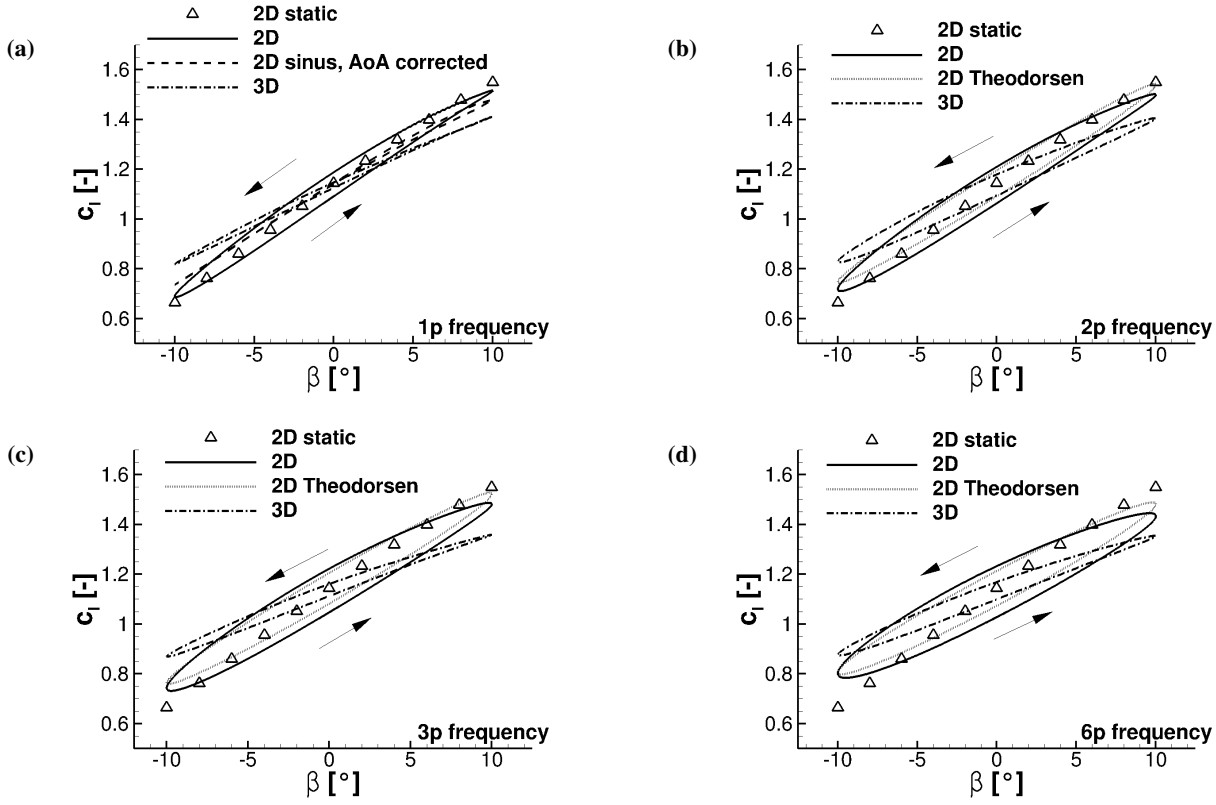

**Figure 20.** Comparison of 2D/3D lift for different flap frequencies: 1p **(a)**, 2p **(b)**, 3p **(c)** and 6p **(d)**

As for the higher frequencies the reduced axial velocity method is not applicable but also as the AoA variations are smaller compared to the 1p case, no AoA corrected curve is plotted. For these cases the results obtained with the 2D theory of Theodorsen have been added for comparison. The plots show the expected decrease of lift amplitude from 2D static over

2D sinusoidal to 3D sinusoidal. For the 1p oscillation the comparison of 2D static and 2D sinusoidal results shows the minor influence of unsteady effects. Even though hysteresis begins to develop in the 2D sinusoidal results, the $c_l$ amplitude reduction is still small. This result corresponds well to the low reduced frequency in this case of 0.024. Larger differences are seen by comparing 2D and 3D results which show the decrease of amplitude caused by trailing and shed vorticity. The reduced amplitude leads to less shed vorticity and thus less hysteresis is apparent compared to the 2D solution.

The AoA corrected 2D curve shows the approximate result for a 2D simulation including an AoA variation in the inflow. The curve demonstrates less hysteresis and a smaller amplitude compared to the baseline progression, which is reasonable since the AoA progression is a feedback of the aerodynamic forcing in the 3D case. Like it was noticed in Sect. 4.2, for low flap frequencies the axial induction is able to react to the instantaneous load and mimics the effects. The smaller slope which is seen in the 3D curve can be explained by the decrease of the gradient $dc_l/d\beta$ caused by trailing vortices in 3D.

In 2D unsteady effects constantly increase with the flap frequency for the regarded cases what corresponds well to the results with Theodorsen's 2D theory. The amplitude of lift oscillation is continuously reducing and more pronounced hysteresis is seen. The results obtained by the Theodorsen's theory are in fair agreement to the CFD results. A more symmetrical hysteresis is apparent compared to the slightly bent CFD curves. Generally, the hysteresis direction is in agreement to the observations made by Troldborg (2005). In 3D, the amplitude is also continuously decreasing but hysteresis effects show no clear trend. A slightly bigger hysteresis is seen for the 2p frequency than for the 3p frequency. A reason for this phenomenon could be a different phase lag in the AoA oscillation resulting from the flap deflection. For clarification, an AoA extraction for unsteady cases would be required.

Table 4 lists amplitudes and Table 5 phase lags of the lift coefficient for the different cases in order to quantify the effects. The values were again obtained by applying a Fast-Fourier transformation on the unsteady lift progression and analyzing the main peak in the results. Additionally results of the 2D theory by Theodorsen described in Sect. 2.3 are shown in the table. The results of this simplified theory are in a very good agreement to the results of 2D simulations with regard to the $c_l$ amplitude. Only at 6p a slight difference is noticeable. Larger differences are observed in the phase shift, for which higher values are determined in the 2D CFD simulation. These differences can be caused by the assumption of thin airfoils in Theodorsen's theory as similar observations were found by Motta et al. (2015) for pitching airfoils.

**Table 4.** $c_l$ amplitude, 75 % blade cut, 2D and 3D results

|  | 1p | 2p | 3p | 6p |
|---|---|---|---|---|
| 2D | 0.42 | 0.40 | 0.38 | 0.33 |
| 3D | 0.30 | 0.29 | 0.25 | 0.24 |
| $\Delta c_{l,3D}/\Delta c_{l,2D}$ | 71% | 73% | 66% | 73% |
| 2D, Theodorsen | 0.42 | 0.40 | 0.38 | 0.35 |

**Table 5.** $c_l$ phase shift, 75 % blade cut, 2D results

|  | 1p | 2p | 3p | 6p |
|---|---|---|---|---|
| 2D | -6.3 | -10.2 | -12.9 | -17.7 |
| 2D, Theodorsen | -4.9 | -7.5 | -9.2 | -11.7 |

The comparison of the 2D and 3D lift amplitudes shows the expected reduction due to the influence of trailing vortices. The 3D lift amplitude alleviates to 66-73 % of the result of 2D simulations at the same flap frequency based on the round values listed in Table 4. As it was noticed before in Sect. 4.1 with regard to the thrust oscillation, the lift amplitude of the 3p and 6p case is similar. This is also seen in the relative amplitude reduction, as 3p shows a significantly lower value than the other cases.
Unfortunately, the reason for this phenomenon could not finally be clarified since this would require a method to extract the AoA in transient cases, too. This would allow to judge and compare the AoA oscillation between all 3D cases, which provides further insight. Further research is required in this respect. Nevertheless, this relative reduction is roughly constant throughout all flap frequencies. An earlier investigation (Jost et al., 2016) focused on the impact of steady flap deflections on the blade performance. Various flap extensions along the blade radius (10 % and 20 %) and chord (10 % and 30 %) were analyzed in a
parametric study at 15 m/s wind speed. The analysis included also a comparison to 2D simulations and an extract of the results is shown in Table 6 for positive and negative deflections.

**Table 6.** $c_l$ amplitude, 75 % blade cut, 15 m/s, steady deflections (Jost et al., 2016)

|  | $\beta = 10°$ | | $\beta = -10°$ | |
|---|---|---|---|---|
|  | 10 % chord | 30 %chord | 10 % chord | 30 %chord |
| $\Delta c_{l,3D,10\%span}/\Delta c_{l,2D}$ | 70% | 69% | 65% | 65% |
| $\Delta c_{l,3D,20\%span}/\Delta c_{l,2D}$ | 80% | 79% | 75% | 77% |

A correlation to the results of the present study can be noticed. The relative amplitude reduction $(\Delta c_{l,3D,10\%span}/\Delta c_{l,2D})$ is very similar for both chord extents and in a comparable range as observed in the oscillating cases. Please note that the slightly lower values for steady deflections can be explained by the adjusting axial induction as described in Sect. 4.2. The
value for 20 % radial extent $(\Delta c_{l,3D,20\%span}/\Delta c_{l,2D})$ is in contrast about 10 % higher. Consequently, the relative lift amplitude reduction at mid flap position $(\Delta c_{l,3D}/\Delta c_{l,2D})_{mid,flap}$ can serve as a rough characteristic value for a certain flap layout. It also allows a decoupled consideration of 3D and unsteady effects at this location. This means that as a first estimation $(\Delta c_{l,3D}/\Delta c_{l,2D})_{mid,flap}$ can be determined based on a 3D simulation with static maximum flap deflection. The amplitude reduction by shed vorticity can be investigated separately in 2D and can then be superimposed with the 3D result. This simpli-
fied or quasi-2D approach with regard to shed vorticity is however only valid at the mid flap area. Closer to the flaps edges the unsteady effects will also become three-dimensional.

With respect to a more general aerodynamic modelling in lower fidelity tools, the present investigation suggests a simplified consideration of near wake effects. The dominant phenomena could be assigned to trailing and shed vorticity which can be captured by such approaches. First comparisons of the present simulations to the near wake model by Pirrung et al. (2016), an implementation in the BEM-code HAWC2 by DTU (Technical University of Denmark), can be found in Barlas et al. (2016). Aparicio et al. (2016a) have also published favorable results following a similar approach in the BEM-code FAST, which was developed by NREL (National Renewable Energy Laboratory). Further benchmarks were performed within the AVATAR project (Manolesos and Prospathopoulos, 2015; Ferreira et al., 2015; Aparicio et al., 2016b).

The results of the moment coefficient $c_m$ are depicted in Fig. 21 for the 1p and 6p frequency. It is noted that while the previous plots of the 3D torsion moment were evaluated relative to the pitch axis, the results have been transferred to the quarter chord point for the 2D comparison. In general, only a minor influence of unsteady aerodynamic effects is observed. A slight reduction of the amplitude and small hysteresis is seen in the 3D case. The 2D results are in a good agreement to (Ferreira et al., 2015), for which similar simulations have been performed for the same airfoil.

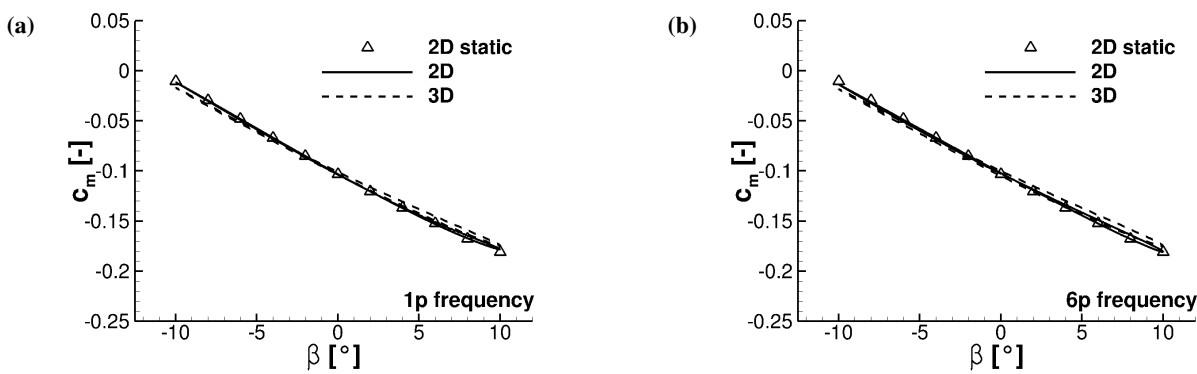

**Figure 21.** Comparison of 2D/3D moment coefficient for different flap frequencies: 1p **(a)**, 6p **(b)**

## 5   Conclusions

In the present work the influence of unsteady 3D effects on trailing edge flaps has been investigated for the case of harmonic oscillations of the flap angle. For this purpose four different flap frequencies ranging from 1p to 6p of the rotational frequency were simulated on a 2D airfoil and 3D rotor. The simulations of the 3D rotor showed a significant influence of trailing and shed vorticity. This leads to a reduction of the amplitude and a phase lag in the load response to the flap signal with increasing flap frequency. This behavior was observed in both, integral and sectionally distributed forces along the blade radius. The negative influence of trailing vortices in the flap section is however partly compensated by an increased efficiency of neighboring sections due to the induced up- or downwash. A high impact of dynamic inflow effects is observed in the comparison to steady flap deflections. Unlike for the oscillating cases, the axial induction of the turbine is able to fully adjust to the changed load situation for steady flap deflections which is noticable in the rotor loads.

A detailed investigation of the 75 % blade cut representing the mid flap position was performed in order to compare 2D airfoil to 3D rotor results and thereby quantify unsteady 2D and 3D effects. The 2D CFD simulations were additionally compared to the simplified model by Theodorsen, a widespread analytical approach for this matters. A good agreement between 2D CFD and the Theodorsen model was observed with regard to the amplitude of the lift oscillations. Higher discrepancies could be noticed in the phase shift which could be assigned to Theodorsen's assumption of thin airfoils. The comparison of the 2D and 3D CFD simulations revealed the significantly higher reduction of the lift amplitude in the 3D rotor case due to trailing vorticity. Only about 70 % of the 2D value for the respective flap frequency is achieved. Additionally, less hysteresis is seen at the rotor blade which is connected to the decrease of lift amplitude. 2D and 3D results regarding the moment coefficient were also compared and exhibited an only minor influence of unsteady effects.

As discussed in the present study, an overall CFD-based assessment of the flap concept requires the incorporation of aero-elasticity. On a flexible blade the flap oscillation will be superimposed with pitching and plunging motions. A careful evaluation of these phenomena in comparison to the present investigation will allow an isolation of the different effects and lead to a deeper insight of the 3D characteristics of trailing edge flaps on a real wind turbine rotor. This is the next step in the ongoing research.

*Acknowledgements.* The authors acknowledge European FP7 project Innwind.EU for funding (Grant agreement No. 308974). This work was partly performed on the Supermuc Cluster (LRZ Munich), the HazelHen and NEC Cluster (HLRS Stuttgart) and the bwUniCluster (framework bwHPC, funding: Ministry of Science, Research and Arts and the Universities of the state Baden-Württemberg).

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
