# Peer review of "An investigation of unsteady 3D effects on trailing edge flaps"

_Wind Energy Science, 2016_

## Referee Comment (RC1) · V.A. Riziotis (Referee) · 12 Jan 2017

General comment:

The paper investigates three dimensional and unsteady effects of a moving trailing edge flap on the aerodynamic loads of DTU 10 MW reference turbine through CFD analysis. Emphasis is given on less understood till now three dimensional effects which are definitely not captured by existing state-of-the-art BEM based aeroelastic tools used in the assessment of load alleviation capabilities of flaps. The work is very interesting and relevant to the research work recently undergone in EU funded projects INNWIND.EU and AVATAR.

Specific comments:

-In section 2, consider moving equation 2 up below equation 1, and add "According to Theodorsen's method the lift is given by:" and explain the terms of the equation afterwards.

-In section 2, I believe it is not mentioned in the text what is the amplitude of the periodic flap motion used in deriving the results of figure 2. An idea could be to plot transfer function results of DCL/Dbeta in figure 2 instead of DCL.

-In section 3.5, second paragraph. It would be instructive for the reader if you describe in detail inflow/operational conditions that lead to the dimensionless parameters used in the 2D analysis. For example inflow velocity, rotational speed (or tip speed ratio), spanwise position of the section.

- In section 4.1, page 10, below line 10, it is the effect of the induced drag that the authors are describing. When the flap angle increases, lift increases so the intensity of the trailed vorticity increases, and therefore downwash and induced drag increases while effective AoA decreases. The opposite happens at negative flap angles. A similar discussion is made on section 4.3. It is again the effect of the induced drag that differentiates the Cdmean over Cd. The comment here is that it seems that discussions in sections 4.1,4.2 and 4.3 are somehow separated although they should be finally linked together. Especially the results of figure 16 and 17 must be linked to the results of figure 10,11 and 12. Also the concept of the induced drag (which is a consequence of course of the downwash effect) could be introduced.

-Also in section 4.3. Why do we need AoA for in a CFD simulation? The only reason why we would like to have a consistent definition of the AoA is in order to be able to compare against BEM or lifting line models and tune them. Perhaps this objective should be pointed out in the text because otherwise the need for paying so much attention on extracting AoA is not clear.

-The term downwind/upwind is used for the effect of induced velocities. Perhaps it would be better to use upwash/downwash.

Technical comments. Some editorial typo/syntax changes,

-Page 2, line 17, "However the blade parts next to the flap….", could be better to replace "next" by "adjacent"?

- Page 2, line 23,"…which in turn influences (in the) blade loads"

-Page 3, line 25, "Both is (are)…"

-Page 7, line 13, "…corresponding to the sixth (six times or sixth harmonic) of the rotational velocity"

-Page 9, line 12, "…not updated (in) every time step…"

-Page 9, line 16, "…correspond to reduced (of) frequencies…"

-Page 9, line 18, "A higher (frequent) frequency …"

-Page 10, below line 5, It is not clear what does the sentence in the parenthesis mean (respectively 0.2 in Fig.10a). Also the next sentence could be re-phrased. In simpler words it is the superposition of two variations with the same frequency by different phases that causes the effect.

Page 10, line 13, "..appear (appearing).."

Page 12, The caption of Table 2 makes no sense. Could be integral quantities for constant flap position.

Page 14, First sentence of section 4.3 could be re-phrased. Moreover you can define influence on what?

Page 14, line 15, "…but (strongly) strong.

Page 14 line 16, "…as driving and thrust force cpomponents.."

Page 16, first sentence, "an FFT"

Page 16, before last sentence. "In the following….", please re-phrase.

Page 16, line 19, "Even though a beginning hysteresis...", consider re-phrasing to "Even though hysteresis begins to develop in 2D..."

Page 18, line 8, "..with regard to..."

Page 19, line 4, "Unlike the oscillating cases..."

Perhaps punctuation should be also checked.
* * *

---

## Referee Comment (RC2) · Anonymous Referee #2 · 22 Jan 2017

This paper presents some CFD results aiming to investigate the effect of trailing edge flaps on the aerodynamics of wind turbines. The paper is at the level of what is expected for a conference presentation but it lacks in originality, and depth to be considered as a journal publication.

To begin with, the paper is not presenting something new in terms of predictive aerodynamic methods, and it is not even applying existing methods in an innovative way.

In addition, the depth of the presented analysis of the effect of trailing edge flaps on the 3D unsteady aerodynamics of a wind turbine is at least superficial. The authors compare 2D and 3D results for the flapped wind turbine section but this comparison is not enough to provide a detailed understanding of how much the trailed vorticity and the additional edge effects of the finite span flap can be quantified in a way that

engineers can use for practical calculations. After all the CFD computations the reader expects some model/table/equation that will be transferable to other cases. Otherwise this work will have to be repeated for any other wind turbine and for any other flap location apart from the selected one here.

More interestingly, the paper is not considering the simple fact that once a flap is deployed, it will begin to affect the sectional pitching moments, it will lead to different displacement and perhaps torsion of the section and this will result in a different behavior than what is shown in the paper. For the large flexible blades of wind turbines, including the effect of aeroelasticity is important and this effect is neglected in this paper.

In conclusion, I feel that the authors are on the right track but they have to investigate thing deeper, collect a volume of work that covers several cases and try to summarise the effects of the flap in a consolidated way in a model. Their work and method are good but they are simply not there yet to be able to provide the depth of analysis required by an archival journal publication.

---

## Author Response (AR1)

The present document includes:

1. Point-by-point response to 1$^{st}$ reviewer
2. Point-by-point response to 2$^{nd}$ reviewer
3. List of all major changes on the manuscript
4. Marked-up manuscript (Sections which include major changes are marked in red colour.)

The authors would like to thank the reviewer for his efforts and valuable comments. They are very much appreciated and will be/are incorporated into the revised paper!

In the present document the comments given by the reviewer are addressed consecutively. The following formatting is chosen:

- The reviewer comments are marked in blue and italic.
- The reply by the authors is in black colour.
- Changed/extracted text sections are in green boxes.

A revised manuscript which incorporates the comments of both reviewers has been prepared. The made changes with respect to the 1st reviewer are listed below. At the end of this document a marked-up manuscript is appended. Sections which include major changes are marked in red colour.

**General comment:**

*"The paper investigates three dimensional and unsteady effects of a moving trailing edge flap on the aerodynamic loads of DTU 10 MW reference turbine through CFD analysis. Emphasis is given on less understood till now three dimensional effects which are definitely not captured by existing state-of-the-art BEM based aeroelastic tools used in the assessment of load alleviation capabilities of flaps. The work is very interesting and relevant to the research work recently undergone in EU funded projects INNWIND.EU and AVATAR."*

Thank you very much!

**Specific comments:**

*-In section 2, consider moving equation 2 up below equation 1, and add "According to Theodorsen's method the lift is given by:" and explain the terms of the equation afterwards.*

Thank you. This has been changed.

*-In section 2, I believe it is not mentioned in the text what is the amplitude of the periodic flap motion used in deriving the results of figure 2. An idea could be to plot transfer function results of DCL/Dbeta in figure 2 instead of DCL.*

Thank you. That's true and was missed. The transfer function $dc_l/d\beta$ is plotted now and the text was slightly changed to account for this (revised manuscript: page 4, line 25):.

> … The instantaneous lift coefficient $c_l(t)$ can be analyzed with regard to the amplitude $\Delta c_l$ and phase shift $\Phi$ of the lift response with respect to the input flap signal. $\Delta c_l$ can also be evaluated in relation to the amplitude of flap deflection ($\Delta c_l/\Delta\beta$). …
>
>
[Figure]

>
> Figure 3. **Lift amplitude in relation to flap amplitude $\Delta c_l/\Delta\beta$ and phase shift according to Theodorsen**

*-In section 3.5, second paragraph. It would be instructive for the reader if you describe in detail inflow/operational conditions that lead to the dimensionless parameters used in the 2D analysis. For example inflow velocity, rotational speed (or tip speed ratio), spanwise position of the section.*

Thank you. A table showing the rated operational conditions has been added and is referenced throughout the text (revised paper: page 8, line 17).

**… The simulations have been performed at a realistic inflow extracted from the 3D rotor case at rated operational conditions. These conditions are specified in Table 2.**

Table 2. **DTU 10 MW turbine, rated operational conditions**

| Wind speed | Rotational speed | Blade pitch | Tip speed ratio |
|------------|------------------|-------------|-----------------|
| 11.4 m/s | 9.6 rpm | 0° | 7.86 |

**At 75% radius, the Reynolds number was determined to 15.4 millions, Mach number to 0.2 and the AoA to 6.5°. …**

*- In section 4.1, page 10, below line 10, it is the effect of the induced drag that the authors are describing. When the flap angle increases, lift increases so the intensity of the trailed vorticity increases, and therefore downwash and induced drag increases while effective AoA decreases. The opposite happens at negative flap angles. A similar discussion is made on section 4.3. It is again the effect of the induced drag that differentiates the Cdmean over Cd. The comment here is that it seems that discussions in sections 4.1,4.2 and 4.3 are somehow separated although they should be finally linked together. Especially the results of figure 16 and 17 must be linked to the results of figure 10,11 and 12. Also the concept of the induced drag (which is a consequence of course of the downwash effect) could be introduced.*

Thank you very much for this valuable comment!

The paragraph in section 4.1 was revised as follows (revised paper: page 12, line 5). Please note that the torsion moment has been added as suggested by the second reviewer.

**In Fig. 12 and Fig. 13 sectional distributions of driving force, thrust and torsion moment over the blade radius are shown for 1p and 6p case respectively. Four instantaneous solutions are plotted for maximum, minimum and 0° flap deflection. Thrust shows the expected increase and decrease in the flap section with a smooth load distribution over the flap edges. This smoothing is a consequence of the positive effect of the flap deflection on neighboring blade sections as described in Sect. 2.2. While trailing vorticity reduces the effect of the flap in the flap section compared to 2D, the sections next to the flap part produce higher/lower lift due to the induced upwash/downwash for respectively positive/negative flap angles. The change of sign in induced velocity caused by the flap edge vortices is also apparent in the driving force as significant steps are appearing at the transition between flap and rigid rotor part. An opposite behavior of sectional driving force in relation to thrust can be noticed by comparing the diagrams. When thrust increases locally in the flap area, the driving force decreases in relation to neighboring sections. This results again from the strong influence of drag on the driving component at rated wind turbine conditions and will be explained on the basis of 1p case as follows. Due to the low reduced frequency in this case (k = 0.024), the influence of shed vorticity is still weak. For maximum positive deflection ($t/T_{Flap}$ = 0.25) the increase of trailing vorticity causes a downwash in the flap section. This reduces the effective AoA and leads to a rise of induced drag in addition to the drag augmentation caused by the flap deflection itself. The overall drag increase is compensated by the lift increase resulting from**

the flap deflection and relative to 0° flap deflection an increase of driving force is achieved. The neighboring sections to the flap experience an additional upwash in case of positive deflections. Consequently, the induced drag reduces associated with the lift increase and these sections produce in total a higher sectional driving force. Similar observations are made vice versa for maximum negative deflections, but the driving force increase in the flap section is less pronounced compared to the decrease in case of positive deflection. Further elaborations in this respect can be found in Sect. 4.3, in which lift and drag forces are extracted and compared. With regard to the torsion moment around the pitch axis, a strong oscillation is seen in flap section with steep gradients at the flap edges. This torsion moment or $c_m$ oscillation is typical for trailing edge flaps (Ferreira et al., 2015) and its effect on the overall performance of the flap concept needs to be investigated separately in an aero-elastic simulation when the blade is able to twist.

[Figure]

Figure 12. **1p sectional forces (driving force (a), thrust (b) and local torsion moment (c))**

[Figure]

Figure 13. **6p sectional forces (driving force (a), thrust (b) and local torsion moment (c))**

The concept of induced drag is introduced in section 2 (revised paper: page 3, line 10):

Generally, the efficiency of the flap with regard to local lift increase or decrease is reduced by trailed vorticity in the 3D case. The flap deflection causes an additional downwash or upwash in the flap section. This leads to a respectively lower or higher effective AoA in the 3D case and consequently to induced drag in relation to the baseline AoA. It is worth noting that with respect to the case without flap deflection (β=0°), the induced drag is increased in case of positive deflections and decreased in case of negative deflections.

The adverse effect of trailing vortices in the flap section is however countered by a positive effect in the blade parts adjacent to the flap section. Caused by the sign change of induced velocities over the flap edge, the described effects for the flap section are experienced vice versa at these blade parts. With regard to integral loads such as power and thrust, this effect opposes the negative impact of trailing vortices in the flap section.

Please note that in the revised manuscript, Sect. 2 is divided in three subsections in connection to the comments by the second reviewer. (1. 2D airfoil, 2. 3D rotor blade, 3. Theodorsen theory)

In section 4.3, bigger modifications have been performed in order to describe the intention of the section more clearly. This is also linked to the next comment by the reviewer:

*-Also in section 4.3. Why do we need AoA for in a CFD simulation? The only reason why we would like to have a consistent definition of the AoA is in order to be able to compare against BEM or lifting line models and tune them. Perhaps this objective should be pointed out in the text because otherwise the need for paying so much attention on extracting AoA is not clear.*

Changed introduction to section 4.3 (revised paper: page 17, line 8):

**4.3 Influence of varying AoA in 3D**
**As observed in the previous section, in the 3D rotor case the local AoA is oscillating over a flap period as a result of dynamic inflow. This means from an aerodynamic point of view that two unsteady mechanisms are superimposed: pitch and flap oscillation. The objective of the present work is though to characterize and quantify unsteady 3D effects solely due to flap deflection and consequently some preliminary considerations have to be made. For this purpose the 1p frequency is regarded in the following, for which quasi-steady assumptions are eligible and the reduced axial velocity method can be applied. The variation of the local inflow velocity and the AoA is shown in Fig. 17 for the mid flap position. While the inflow velocity shows no major variations, the AoA oscillates with an amplitude of 0.6°.**

[Figure]

Figure 17. **1p instantaneous inflow conditions 3D, 75 % radius**

**When the instantaneous AoA is extracted from the 3D simulation it includes the oscillations caused by both mechanisms, dynamic inflow and flap oscillation. The dynamic inflow oscillation represents an oscillation of the baseline AoA as a result from the variation of the axial induction of the turbine. In contrast the oscillations caused by the flap originate from the downwash of 3D trailing vorticity which changes the effective AoA. As the objective is to quantify 3D trailing vorticity, the flap-caused AoA oscillation should be mimicked to the aerodynamic coefficients, while in theory the influence of the dynamic inflow caused oscillation should be eliminated. A clear distinction between both oscillations is however not possible and requires further aerodynamic modelling. Nevertheless, the influence of the overall AoA oscillation on the 3D extracted aerodynamic coefficients can be assessed.**

Link to section 4.1 and the sectional distribution of driving force and thrust and changed summary of section 4.3 (page 18, line 17):

[Figure]

Figure 18. **1p comparison of lift and drag, 3D instantaneous/ 3D mean, 75 % radius**

**… Consequently, the results for $c_l$ and $c_{l,mean}$ are very similar. $c_d$ and $c_{d,mean}$ differ however strongly. Based on the previous considerations this difference consists of two parts, induced drag which originates from trailed vorticity and the drag resulting from the AoA oscillation caused by dynamic inflow.**
**The plot in Fig. 18 can be directly linked to the curves for driving force and thrust in the 1p case displayed Fig. 11 (in original manuscript: Fig. 10) and Fig. 12 (in original manuscript: Fig. 11). Thrust is dominated by $c_l$ and consequently the progressions over a flap period are very similar. The driving component is a superposition of both forces which are oscillating at a different phase. This can be noticed for example at the time instance when the flap is deployed to maximum deflection ($t/T_{Flap}$ = 0.25). $c_l$ but also $c_{d,mean}$ is maximal and as a result the progression of driving force flattens. Vice versa the phenomenon is observed at minimum deflection ($t/T_{Flap}$ = 0.75).**
**With regard to the objective of this section, determining the influence of the AoA oscillation on the 3D extracted aerodynamic coefficients, it can be concluded that the extracted $c_l$ is only minor affected and it is eligible to use $c_{l,mean}$ for the comparison to 2D simulations and the evaluation of the impact of trailing vortices. With respect to $c_d$ it is difficult to clearly distinguish between the part caused by the AoA oscillation and the induced drag from trailed vorticity. In order to judge the impact of trailing vortices on $c_d$, this differentiation is however necessary and the part caused by the AoA oscillation needs to be eliminated. The emphasis of the comparison to 2D simulations is hence on lift and moment coefficient.**

*-The term downwind/upwind is used for the effect of induced velocities. Perhaps it would be better to use upwash/downwash.*

Thank you. The authors agree and this has been changed through out the text.

In addition to the above changes, two smaller corrections had to be performed by authors for correctness:

- Table 2, original manuscript: An error was found in post-processing for β=10° (evaluation of wrong revolution) which changed the results to a power of 10.71MW (reduction by 0.37%) and to a thrust of 1790kN (reduction by 0.17%). The conclusions are unaffected.
- Table 3, original manuscript: corrected typo for (Δcl,3D /Δcl,2D) to 73%

**Technical comments. Some editorial typo/syntax changes,**

*-Page 2, line 17, "However the blade parts next to the flap. . ..", could be better to replace "next" by "adjacent"?*

*- Page 2, line 23,". . .which in turn influences (in the) blade loads"*

*-Page 3, line 25, "Both is (are). . ."*

Thank you. All of the above comments have been changed as specified.

*-Page 7, line 13, ". . .corresponding to the sixth (six times or sixth harmonic) of the rotational velocity"*

Thank you. This has been changed to:

> **…corresponding to six times the rotational frequency…**

*-Page 9, line 12, ". . .not updated (in) every time step. . ."*

*-Page 9, line 16, ". . .correspond to reduced (of) frequencies. . ."*

*-Page 9, line 18, "A higher (frequent) frequency . . ."*

Thank you. All of the above comments have been changed as specified.

*Page 10, below line 5, It is not clear what does the sentence in the parenthesis mean (respectively 0.2 in Fig.10a). Also the next sentence could be re-phrased. In simpler words it is the superposition of two variations with the same frequency by different phases that causes the effect.*

Thank you. The sentences are re-phrased as follows (revised manuscript: page 11, line 26):

> **… For the 3p and 6p case, a second superimposed oscillation is visible from $t/T_{Flap}$= 0-0.2 and $t/T_{Flap}$=0.8-1. This oscillation results from the overlay of lift and drag forces in the rotor plane. At higher frequencies drag shows a significant amplitude increase as seen in Fig. 6 and, additionally, $c_l$ and $c_d$ are oscillating with different phases. The superposition of both force components leads to the curve progression seen in the 3p and 6p case. This phenomenon in driving force is especially present at operational conditions with a pitch angle of 0°, for which the impact of drag is high. …**
>
>
[Figure]

>
> Figure 11. **Variation of flap frequency, 75 % blade cut, driving force (a), thrust (b) and local torsion moment (c)**

*Page 10, line 13, "..appear (appearing).."*

Thank you. The sentences are re-phrased as follows (revised manuscript: page 12, line 10):

> **… The change of sign in induced velocity caused by the flap edge vortices is also apparent in the driving force as significant steps are appearing at the transition between flap and rigid**

**rotor part. …**

*Page 12, The caption of Table 2 makes no sense. Could be integral quantities for constant flap position.*

Thank you, we missed a word here. Additionally, the authors suggest to use normalized integral loads with the value for β=0° to highlight the differences (revised manuscript: page 15, line 5).

**… Like for the oscillating flap, cases the simulations were initiated in steady state with 16000 iterations and then restarted in unsteady mode for three turbine revolutions with a time step corresponding to 2° azimuth. This approach is plausible because the flap area is characterized by a steady flow situation as shown in similar studies (Aparicio et al., 2016b). Table 3 shows the mean integral loads of the third rotor revolution normalized with the respective value for β=0° for a relative comparison.**

Table 3. **Integral loads for steady deflection normalized with value for β=0**

|  | $\beta = 10°$ | $\beta = -10°$ |
|---|---|---|
| Normalized power [%] | 96.2 | 98.3 |
| Normalized thrust [%] | 103 | 95.2 |

*Page 14, First sentence of section 4.3 could be re-phrased. Moreover you can define influence on what?*

Thank you. This passage was re-phrased in connection to the specific comment on that section.

*Page 14, line 15, ". . .but (strongly) strong.*

*Page 14 line 16, ". . .as driving and thrust force components.."*

*Page 16, first sentence, "an FFT"*

Thank you. All of the above comments have been changed as specified.

*Page 16, before last sentence. "In the following. . ..", please re-phrase.*

Thank you. This sentence has been removed in connection to the specific comment on that section.

*Page 16, line 19, "Even though a beginning hysteresis. . .", consider re-phrasing to "Even though hysteresis begins to develop in 2D. . ."*

*Page 18, line 8, "..with regard to. . ."*

Thank you. All of the above comments have been changed as specified.

*Page 19, line 4, "Unlike the oscillating cases. . ."*

Thank you. The sentences are re-phrased as follows (revised manuscript: page 23, line 21):

**Unlike for the oscillating cases…**

*Perhaps punctuation should be also checked.*

Thank you. The text was revised in this respect.

The authors thank the reviewer for his/her work and respect his/her opinion.

In the present document the comments given by the reviewer are addressed consecutively. The following formatting is chosen:

- The reviewer comments are marked in blue and italic.
- The reply by the authors is in black colour.
- Changed/extracted text sections are in green boxes.

A revised manuscript which incorporates the comments of both reviewers has been prepared. The made changes with respect to the 2nd reviewer are listed below. At the end of this document a marked-up manuscript is appended. Sections which include major changes are marked in red colour.

*"This paper presents some CFD results aiming to investigate the effect of trailing edge flaps on the aerodynamics of wind turbines. The paper is at the level of what is expected for a conference presentation but it lacks in originality, and depth to be considered as a journal publication.*

*To begin with, the paper is not presenting something new in terms of predictive aerodynamic methods, and it is not even applying existing methods in an innovative way. "*

Previously published work on trailing edge flaps was in the large majority performed on the basis of blade element momentum methods, only very little is available on the basis of vortex models and CFD simulations were mostly performed on 2D airfoils. Only few papers are available for the 3D rotor which mainly addressed different objectives. This motivated the present 3D CFD studies.

In order to document this clearly, the introduction (revised manuscript: page 1, line 21) was revised:

> **Over the last years, several investigations showed the potential of the flap concept as for example a test on a full-scale turbine performed by Castaignet et al. (2014). In aero-elastic simulations, fatigue load reductions up to approximately 30 % have been found for a trailing edge flap covering up to 25 % of the blade span of a 5 MW turbine (Barlas et al., 2012a). In most of the numerical studies the aerodynamic loading was computed by blade element momentum (BEM) codes (e.g. Bernhammer et al. (2016), Chen et al. (2017), Ungurán and Kühn (2016)), which have been extended with different engineering models to account for the unsteady flow (e.g. Bergami and Gaunaa (2012)). As viscous and unsteady aerodynamics have a great influence on dynamically deflected flaps (Leishman, 1994), it is however important to also apply higher fidelity models and gain knowledge of the flow physics. In this respect, a lot of studies have been performed on 2D airfoils, for example by Troldborg (2005) and Wolff et al. (2014) using CFD. 2D comparisons of simulation methods with different aerodynamic fidelities were performed by Bergami et al. (2015). For the 3D wind turbine rotor only few publications based on higher fidelity aerodynamic models are available. In 2012 Barlas et al. (2012b) compared CFD to BEM predictions for a rotor with trailing edge flap in an artificial half-wake scenario and found a reasonably good agreement with regard to the complexity of the test case. Leble et al. (2015) investigated trailing edge flaps on the 3D rotor as part of the European AVATAR project and proved the load alleviation potential using a CFD approach. Several comparisons of codes with different aerodynamic fidelities can also be found in the AVATAR project reports (Manolesos and Prospathopoulus (2015), Ferreira et al. (2015), Aparicio et al. (2016b)).. A benchmarking within the European Innwind.EU project (Jost et al. (2015a), Barlas et al. (2016)) showed however that there are still differences between the results of CFD simulations and BEM methods which need to be analyzed. While a previous investigation focused on the analysis of static flap deflection angles (Jost et al., 2016) by means of CFD, the main objective of the present work is to study the influence of unsteady 3D effects on the**

**example of harmonically oscillating morphing flaps.**

The intention of the present paper is to demonstrate that the deflection of the flap on the 3D rotor causes a complex wake development and induction which influences the aerodynamic loads over large parts of the blade. This knowledge is essential for the development of smart rotors and also improved wake models. As a matter of fact the present work and similar simulations were used for the verification of aerodynamic engineering models in BEM within the European projects Innwind.EU and AVATAR. In this background the authors are not aware of any similar study for large wind turbines, but would appreciate any indications.

*"In addition, the depth of the presented analysis of the effect of trailing edge flaps on the 3D unsteady aerodynamics of a wind turbine is at least superficial. The authors compare 2D and 3D results for the flapped wind turbine section but this comparison is not enough to provide a detailed understanding of how much the trailed vorticity and the additional edge effects of the finite span flap can be quantified in a way that engineers can use for practical calculations. "*

We agree with the remark that further studies are required to quantify the effect of trailed and shed vorticity in more practical way. The objective of the present work is to investigate the underlying flow phenomena, identify the dominant aerodynamic effects and consequently provide the basis to derive such models. In this respect, Sect 4.4 was revised to clarify the content (revised manuscript: page 22, line 1).

The comparison of the 2D and 3D lift amplitudes shows the expected reduction due to the influence of trailing vortices. The 3D lift amplitude alleviates to 66-73 % of the result of 2D simulations at the same flap frequency based on the round values listed in Table 4. As it was noticed before in Sect. 4.1 with regard to the thrust oscillation, the lift amplitude of the 3p and 6p case is similar. This is also seen in the relative amplitude reduction, as 3p shows a significantly lower value than the other cases. Unfortunately, the reason for this phenomenon could not finally be clarified since this would require a method to extract the AoA in transient cases, too. This would allow to judge and compare the AoA oscillation between all 3D cases, which provides further insight. Further research is required in this respect. Nevertheless, this relative reduction is roughly constant throughout all flap frequencies. An earlier investigation (Jost et al., 2016) focused on the impact of steady flap deflections on the blade performance. Various flap extensions along the blade radius (10 % and 20 %) and chord (10 % and 30 %) were analyzed in a parametric study at 15 m/s wind speed. The analysis included also a comparison to 2D simulations and an extract of the results is shown in Table 6 for positive and negative deflections.

Table 6. $c_l$ amplitude, 75 % blade cut, 15 m/s, steady deflections (Jost et al., 2016)

|  | $\beta = 10°$ | | $\beta = -10°$ | |
|---|---|---|---|---|
|  | 10% chord | 30%chord | 10% chord | 30%chord |
| $\Delta c_{l,3D,10\%span}/\Delta c_{l,2D}$ | 70% | 69% | 65% | 65% |
| $\Delta c_{l,3D,20\%span}/\Delta c_{l,2D}$ | 80% | 79% | 75% | 77% |

A correlation to the results of the present study can be noticed. The relative amplitude reduction ($\Delta c_{l,3D,10\%span}/\Delta c_{l,2D}$) is very similar for both chord extents and in a comparable range as observed in the oscillating cases. Please note that the slightly lower values for steady deflections can be explained by the adjusting axial induction as described in Sect. 4.2. The value for 20 % radial extent ($\Delta c_{l,3D,20\%span}/\Delta c_{l,2D}$) is in contrast about 10% higher. Consequently, the relative lift amplitude reduction at mid flap position ($\Delta c_{l,3D}/\Delta c_{l,2D})_{mid,flap}$ can serve as a rough characteristic value for a certain flap layout. It also allows a decoupled consideration of 3D and unsteady effects at this location. This means that as a first estimation ($\Delta c_{l,3D}/\Delta c_{l,2D})_{mid,flap}$ can be determined based on a 3D simulation with static maximum flap deflection. The amplitude reduction by shed vorticity can be investigated separately in 2D and can then be superimposed with the 3D result. This simplified or quasi-2D approach with regard to shed vorticity is however only valid at the mid flap area. Closer to the flaps edges the unsteady effects will also become three-dimensional.

*"After all the CFD computations the reader expects some model/table/equation that will be transferable to other cases. Otherwise this work will have to be repeated for any other wind turbine and for any other flap location apart from the selected one here."*

With regard to an analytical model description, the CFD results suggest the consideration of trailing and shed vorticity in the turbine near wake. This approach is followed by several researchers, e.g.:

- M. Aparicio et al. ( http://dx.doi.org/10.1088/1742-6596/753/8/082001 )
- G. Pirrung et al. ( http://dx.doi.org/10.1002/we.1969 )

A first comparison to the present work is performed in:

- Barlas et al. (http://doi.org/10.1088/1742-6596/753/2/022027 )

Similar comparisons were performed in the AVATAR project and can be found in:

- Deliverables of work package 3 (D3.1, D3.2, D3.3),
  (www.eera-avatar.eu/publications-results-and-links)

The authors agree that this was not accurately documented in the original manuscript. The following text is added to section 4.4 (revised manuscript: page 23, line 1):

> **With respect to a more general aerodynamic modelling in lower fidelity tools, the present investigation suggests a simplified consideration of near wake effects. The dominant phenomena could be assigned to trailing and shed vorticity which can be captured by such approaches. First comparisons of the present simulations to the near wake model by Pirrung et al. (2016), an implementation in the BEM-code HAWC2 by DTU (Technical University of Denmark), can be found in (Barlas et al., 2016). Aparicio et al. (2016a) have also published favorable results following a similar approach in the BEM-code FAST, which was developed by NREL (National Renewable Energy Laboratory). Further benchmarks were performed within the AVATAR project (Manolesos and Prospathopoulus (2015), Ferreira et al. (2015), Aparicio et al. (2016b)).**

> *"More interestingly, the paper is not considering the simple fact that once a flap is deployed, it will begin to affect the sectional pitching moments, it will lead to different displacement and perhaps torsion of the section and this will result in a different behavior than what is shown in the paper. For the large flexible blades of wind turbines, including the effect of aeroelasticity is important and this effect is neglected in this paper."*

The authors agree that the pitching moment and aero-elasticity is important when trailing edge flaps are applied on a wind turbine rotor. Especially when it comes to assessing the overall performance of the flap concept, this is essential.

The focus of the present work is to study the aerodynamic effects of the flap and provide a basis for the derivation of simplified engineering models. In this respect, aero-elasticity is deliberately neglected to capture solely flap effects. As written by the reviewer, in an aero-elastic simulation the blade flapping and torsion would lead to an overlapped plunging and pitching motion of the airfoil. A distinction between the different aerodynamic effects is then difficult.

In order clarify the intentions of the present work, the following has been added to the end of Sect. 1 (revised manuscript: page 2, line 13):

> **It shall be noted that the present work does not aim towards an assessment of the flap concept. The objective is to investigate unsteady 3D aerodynamic effects caused by trailing edge flaps and to obtain deeper knowledge about the dominant phenomena as fundamental basis for an enhancement of engineering tools commonly used for load calculations. Within**

> **this respect aero-elasticity is not considered since on a flexible blade pitching and plunging movements are superimposed to the flap oscillation and a distinction of the isolated effects is difficult.**

Additionally an outlook is added to the conclusion (Sect. 5, revised manuscript: page 24, line 10) :

> **As discussed in the present study, an overall CFD-based assessment of the flap concept requires the incorporation of aero-elasticity. On a flexible blade the flap oscillation will be superimposed with pitching and plunging motions. A careful evaluation of these phenomena in comparison to the present investigation will allow an isolation of the different effects and lead to a deeper insight of the 3D characteristics of trailing edge flaps on a real wind turbine rotor. This is the next step in the ongoing research.**

The evaluation of the pitching moment is added to the paper. For this, the following modifications have been made to the revised manuscript:

- Section 2.1 is added to introduce the aerodynamic effects of the flap on $c_l$, $c_d$ and $c_m$. As by this section 2 has become consequently more elaborate, it is divided in three subsections. (1. 2D airfoil, 2. 3D rotor blade, 3. Theodorsen theory) (revised manuscript: page 2, line 24)

[revised manuscript text omitted]

*"In conclusion, I feel that the authors are on the right track but they have to investigate thing deeper, collect a volume of work that covers several cases and try to summarise the effects of the flap in a consolidated way in a model. Their work and method are good but they are simply not there yet to be able to provide the depth of analysis required by an archival journal publication."*

Thank you. The authors aim towards a deeper investigation in the revised manuscript. Exemplary two extracts are shown here.

- Section 4.1: Elaboration with regard to sectional force distributions (revised manuscript: page 12, line 6)

In Fig. 12 and Fig. 13 sectional distributions of driving force, thrust and torsion moment over the blade radius are shown for 1p and 6p case respectively. Four instantaneous solutions are plotted for maximum, minimum and 0° flap deflection. Thrust shows the expected increase and decrease in the flap section with a smooth load distribution over the flap edges. This smoothing is a consequence of the positive effect of the flap deflection on neighboring blade sections as described in Sect. 2.2. While trailing vorticity reduces the effect of the flap in the flap section compared to 2D, the sections next to the flap part produce higher/lower lift due to the induced upwash/downwash for respectively positive/negative flap angles. The change of sign in induced velocity caused by the flap edge vortices is also apparent in the driving force as significant steps are appearing at the transition between flap and rigid rotor part. An opposite behavior of sectional driving force in relation to thrust can be noticed by comparing the diagrams. When thrust increases locally in the flap area, the driving force decreases in relation to neighboring sections. This results again from the strong influence of drag on the driving component at rated wind turbine conditions and will be explained on the basis of 1p case as follows. Due to the low reduced frequency in this case ($k = 0.024$), the influence of shed vorticity is still weak. For maximum positive deflection ($t/T_{Flap} = 0.25$) the increase of trailing vorticity causes a downwash in the flap section. This reduces the effective AoA and leads to a rise of induced drag in addition to the drag augmentation caused by the flap deflection itself. The overall drag increase is compensated by the lift increase resulting from the flap deflection and relative to 0° flap deflection an increase of driving force is achieved. The neighboring sections to the flap experience an additional upwash in case of positive deflections. Consequently, the induced drag reduces associated with the lift increase and these sections produce in total a higher sectional driving force. Similar observations are made vice versa for maximum negative deflections, but the driving force increase in the flap section is less pronounced compared to the decrease in case of positive deflection. Further elaborations in this respect can be found in Sect. 4.3, in which lift and drag forces are extracted and compared. With regard to the torsion moment around the pitch axis, a strong oscillation is seen in flap section with steep gradients at the flap edges. This torsion moment or $c_m$ oscillation is typical for trailing edge flaps (Ferreira et al., 2015) and its effect on the overall performance of the flap concept needs to be investigated separately in an aero-elastic simulation when the blade is able to twist.

[revised manuscript text omitted]

In addition to the above changes, two smaller corrections had to be performed by authors for correctness:

- Table 2, original manuscript: An error was found in post-processing for β=10° (evaluation of wrong revolution) which changed the results to a power of 10.71MW (reduction by 0.37%) and to a thrust of 1790kN (reduction by 0.17%). The conclusions are unaffected.
- Table 3, original manuscript: corrected typo for ($\Delta$cl,3D /$\Delta$cl,2D) to 73%

List of all major changes on the manuscript:

- Abstract: page 1, line 5 – 11
- 1. Introduction:
    - p. 1, l. 24 – p. 2, l. 11
    - p. 2, l. 13 – p. 2, l. 22
- 2. Aerodynamic effects of trailing edge flaps
    - p. 2, l. 24 – p. 3, l. 3
    - p. 3, l. 10 – p. 4, l.  3
    - p. 4, equation moved
    - p. 4, l. 24
    - p. 5., l. 7 – 8
- 3. Numerical process chain
    - p. 8, l. 17 – 18, new Table2
    - p. 8, l. 22
    - p. 9, l. 5 – 7
    - p. 9, l. 17 – 21
    - p. 10, Figure 8 caption appended
- 4. Results
    - p. 11, l. 22 – p. 13, l. 13
    - p. 13, Figure 11 caption appended
    - p. 14, Figure 12 and 13 caption appended
    - p. 15, l. 5 – 9, changed Table 3
    - p. 17, l. 9 – p. 18, l. 8
    - p. 18, l. 10 – 12
    - p. 18, l. 17 – p. 19, l. 8
    - p. 21, Table 4 changed
    - p. 22, l. 1 – p.23, l. 12
    - p. 22, new Table 6
    - p. 23, new Figure 21
- 5. Conclusion
    - p. 24, l.8 – 14
- Several references have been added.

[revised manuscript text omitted]

---

## Referee Report (RR1)

[referee-annotated manuscript omitted]